

# Convective distribution of dust over the Arabian Peninsula: the
# impact of model resolution
Jennie Bukowski[1], Susan C. van den Heever[1]
[1] Department of Atmospheric Science, Colorado State University, Fort Collins, CO
*Correspondence to*: Jennie Bukowski (jennie.bukowski@colostate.edu)
**Abstract**
Along the coasts of the Arabian Peninsula, convective dust storms are a considerable source of mineral dust to the
atmosphere. Reliable predictions of convective dust events are necessary to determine their effects on air quality,
visibility, and the radiation budget. In this study, the Weather Research and Forecasting Model coupled with
Chemistry (WRF-Chem) is used to simulate a 2016 summertime dust event over the Arabian Peninsula and examine
the variability in dust fields and associated vertical transport due to the choice of convective parameterization and
explicit versus parameterized convection. Simulations are run at 45 km and 15 km grid spacing with multiple
cumulus parameterizations, and are compared to a 3 km simulation that permits explicit convective processes. Five
separate cumulus parameterizations at 15 km grid spacing were tested to quantify the spread across different
parameterizations. Finally, the impact these variations have on radiation, specifically aerosol heating rates is also
investigated.
On average, in these simulations the explicit case produces higher quantities of dust than the parameterized cases in
terms of dust uplift potential, vertical dust concentrations, and vertical dust fluxes. Major drivers of this discrepancy
between the simulations stem from the explicit case exhibiting higher surface windspeeds during convective activity,
lower dust emission wind threshold velocities due to drier soil, and more frequent, stronger vertical velocities which
transport dust aloft and increase the atmospheric lifetime of these particles. For aerosol heating rates in the lowest
levels, the shortwave effect prevails in the explicit case with a net cooling effect, whereas a longwave net warming
effect is present in the parameterized cases. The spread in dust concentrations across cumulus parameterizations at
the same grid resolution (15 km) is an order of magnitude lower than the impact of moving from parameterized to
explicit convection. We conclude that tuning dust emissions in coarse resolution simulations can only improve the
results to first-order and cannot fully rectify the discrepancies originating from disparities in the representation of
convective dust transport.
**1) Introduction**
Airborne mineral dust is an important atmospheric aerosol (Zender et al., 2004; Ginoux et al., 2012): dust reduces
visibility (e.g. Mahowald et al., 2007; Baddock et al., 2014; Camino et al., 2015) and is detrimental to the human
respiratory system (Prospero, 1999; van Donkelaar et al., 2010; Stafoggia et al., 2016), but also plays a vital role in



fertilizing iron-deficient maritime ecosystems (Martin, 1991; Bishop et al., 2002; Mahowald et al., 2005; Jickells
and Moore, 2015). Dust particles function as cloud condensation nuclei (e.g. Lee et al., 2009; Manktelow et al.,
2009; Twohy et al., 2009; Karydis et al., 2011) and ice nuclei (e.g. DeMott et al., 2003; Field et al., 2006; Knopf and
Koop, 2006; Boose et al., 2016), thereby altering cloud development and properties. Furthermore, mineral dust is of
interest due to its distinctive optical properties; dust both scatters and absorbs shortwave and longwave radiation
(e.g. Tegen et al., 1996; Kinne et al., 2003; Dubovik et al., 2006) modifying atmospheric thermodynamics and the
earth's energy budget in the process (e.g. Slingo et al., 2006; Sokolik and Toon, 2006; Heald et al., 2014).
The influence of atmospheric mineral dust is widespread in the weather and climate system, yet generating skillful
forecasts of dust concentrations and their temporal and spatial evolution has been difficult to achieve. Several
studies suggest that including the radiative effects of mineral dust in numerical weather prediction (NWP) could
refine the radiation balance of these models and improve forecasts (Kischa et al., 2003; Haywood et al., 2005; Pérez
et al., 2006). Advances in climate models have been made by incorporating time-varying dust sources and climate-
dust feedbacks in the radiative forcing calculations (Kok et al., 2014; Woodage and Woodward, 2014; Kok et al.,
2018). However, these potential improvements are contingent upon ingesting both accurate vertical dust
concentrations from models or observations at simulation initialization, as well as correctly representing the coupled
radiative effect dust has on the atmosphere. Still, substantial discrepancies exist between global (Huneeus et al.,
2011) and regional (Uno et al., 2006; Todd et al., 2008) models in the magnitude of predicted dust flux from the
surface to the atmosphere, as well as the models' overall representation of the dust cycle..
A major challenge in accurately modeling dust processes is the scales of motion involved in its emission and
subsequent transport. Dust particles mobilize from the surface due to wind erosion of arid soils, a mechanism that
occurs on the micron scale and must be parameterized in numerical models. Once airborne, mineral dust can deposit
locally or be transported on the synoptic to global scales. Dust events initiate from both large-scale and synoptic
dynamical flow regimes, as well as mesoscale features. Additionally, monsoon circulations (e.g. Marsham et al.,
2008), basin-scale pressure gradients such as the Shamal winds (e.g. Yu et al., 2015), and frontal boundaries (e.g.
Beegum et al., 2018) will produce winds strong enough to emit dust from the surface. Convective outflow
boundaries, also known as haboobs, are an important source of dust to the atmosphere (e.g. Miller et al., 2008), as is
the early morning windspeed maximum resulting from mixing of nocturnal low-level jets (NLLJ) to the surface (e.g.
Fiedler et al., 2013). Wind is the main driver of dust emissions, meaning that the underlying processes that
contribute to the wind fields must be resolved in a model to create an accurate dust forecast.
One potential source of disagreement in models stems from the scaling emissions in dust parameterizations, which
relate the surface emissions proportionally to the second or third power of surface windspeed. This means that minor
miscalculations in modeled windspeeds go on to produce more substantial errors in the dust concentration
calculations (e.g. Menut, 2008). Current aerosol forecast and climate models are run at fine enough grid-spacing to
simulate synoptic events but still typically employ cumulus parameterizations, which are incapable of resolving
many of the mesoscale convective processes which potentially loft or scavenge airborne dust. Pope et al. (2016) and



Largeron et al. (2015) both postulated that this inadequate representation of convection in coarse model simulations,
specifically the underestimation of high surface windspeeds in mesoscale haboobs, is a major contributor to errors in
dust models.
The misrepresentation of dust concentrations in models with cumulus parameterizations has been investigated across
several modeling platforms, mostly from the perspective of dust lofting mechanisms at the surface. Heinold et al.
(2013) ran the UK Met Office Unified Model (UM) over West Africa and found that out of the factors they tested,
the model was most sensitive to explicit versus parameterized convection. Furthermore, in the Heinhold et al. (2013)
study, dust emissions were reduced as grid resolution was increased to convection-permitting scales by roughly
50%. This was found to be due to the parameterized simulations underestimating moist convective activity but
drastically overestimating the NLLJ dust uplift mechanism, a similar relationship to that originally identified in
Marsham et al. (2011).
Conversely, studies using different numerical dust models have identified other relationships between horizontal
resolution and dust emissions. Reinfried et al. (2009) simulated a haboob case study from Morocco with the Lokal
Modell - MultiScale chemistry aerosol transport (LM-MUSCAT, since renamed COSMO-MUSCAT) regional
model and found increased dust emissions in an explicit convection simulation versus those with cumulus
parameterizations. They also established that the model was more sensitive to the choice of cumulus
parameterization rather than the change in horizontal resolution. Similarly, Bouet et al. (2012) identified an increase
in dust emissions with increasing model resolution using the Regional Atmospheric Modeling System coupled to the
Dust Prediction Model (RAMS-DPM) while simulating a Bodélé depression case study. Ridley et al. (2013) showed
that global aerosol models with parameterized convection were also sensitive to model resolution and that higher
horizontal resolution led to higher dust emissions.
With the added computational expense of running aerosol code, the resolution of dust forecast models lags relative
to their weather-only prediction counterparts for both global and regional prediction systems (Benedetti et al., 2014;
Benedetti et al., 2018). Efforts have been made to advance and evaluate predictive aerosol models and ensemble
aerosol modeling with working groups like the International Cooperative for Aerosol Prediction (ICAP) (Benedetti
et al., 2011; Reid, 2011; Sessions et al., 2015), and daily dust forecasts from several aerosol models are now
available through the World Meteorological Organization (WMO) Sand and Dust Storm Warning Advisory and
Assessment System (SDS-WAS) (http://www.wmo.int/sdswas). Nevertheless, none of the modeling groups in the
SDS-WAS currently run at fine enough grid-spacing to explicitly resolve convection (SDS-WAS Model inter-
comparison and forecast evaluation technical manual; last updated January, 2018). While regional numerical
weather prediction models have moved into convection-permitting scales, the added computational cost of aerosol
parameterizations means that convective parameterizations will be a necessity for longer in models that employ
online aerosol predictions. It is also clear that horizontal model resolution, be it specifically as to whether the grid-
spacing is fine enough to permit the explicit resolution of convective processes or is coarse enough to mandate
parameterized convection, is also still an understudied factor in regional dust modeling. As such, exploring





differences across cumulus parameterizations and those relative to convection-permitting resolutions remains
relevant and vital to better understand aerosol forecasting and aerosol-cloud-environment interactions.
While previous studies have begun to examine the effect of horizontal model resolution on dust emissions and
airborne dust concentrations, there are several factors that warrant more investigation. As it stands, there is little
agreement on the sign of the response in dust emissions to a change in model resolution, which seems to vary based
on the regional model being utilized. Most studies have concentrated on the change in dust emissions based on
moving from parameterized convection to explicit convection, while ignoring the possible sensitivity due to the
choice of the cumulus parameterization itself. Furthermore, much of the previous literature focused on how the
increase in resolution affects convective outflow boundaries and surface / near-surface processes as dust sources,
rather than convective transport and the vertical redistribution of dust and its radiative effects at different levels of
the atmosphere. In this paper, we seek to address these limitations in the understanding of the effects of horizontal
model resolution on dust concentrations. The goal of the research presented here is therefore to quantify the sign and
magnitude in the response of modeled dust fields in a regional numerical model to increasing horizontal resolution.
In order to achieve our stated goal, we will use numerical simulations of a case study to examine the variability in
dust emissions and vertical dust concentrations and fluxes due to (1) the choice of convective parameterization, (2)
explicit versus parameterized convection, and (3) the impact of these variations on radiation, specifically aerosol
heating rates. These simulations are performed using the Weather Research and Forecasting Model coupled with
Atmospheric Chemistry (WRF-Chem)  (Skamarock et al., 2008; Grell et al., 2005; Fast et al., 2006) a platform that
has been tested for its sensitivity to vertical resolution for dust extinction coefficient profiles (Teixeira et al., 2015)
and horizontal model resolution and convective transport for chemical species such as carbon monoxide (e.g. Klich
and Fuelberg, 2013), but not for dust. These simulations will represent a case study of a summertime coastal
convective dust event over the Arabian Peninsula, a relatively understudied region compared to areas such as the
Sahara (Jish Prakash et al., 2015), despite being the world's second largest dust emission region (Tanaka and Chiba,

125   2004).

This paper is part of a larger body of collaborative work conducted by the Holistic Analysis of Aerosols in Littoral
Environments (HAALE) research team under the Office of Naval Research Multidisciplinary Research Program of
the University Research Initiative (MURI). The primary goal of the HAALE-MURI project is to isolate the
fundamental environmental factors that govern the spatial distribution and optical properties of littoral zone aerosols.
The study discussed in this manuscript focuses on advancing our understanding in the role that convection plays in
the redistribution of dust aerosol and its radiative effects along the coast of arid regions, and seeks to quantify the
uncertainty in forecasted dust distributions stemming from the representation of convective processes in a regional
model.
The manuscript is organized as follows: an overview of the case study is found in Sect. 2.1, followed by the WRF-
Chem model and physics setup (Sect. 2.2), dust model setup (Sect. 2.3), information about the cumulus
parameterizations and model resolution (Sect. 2.4), and analysis methods in Sect. 2.5. The results are outlined in



Sect. 3, with a discussion on the temporal evolution of dust concentrations and dust uplift potential in Sect. 3.1,
vertical distributions and fluxes of dust in Sect. 3.2, and the effect on aerosol radiative heating rates in Sect. 3.3. A
discussion of the results and implications for the community are located in Sect. 4 and a summary of the findings of
this study are reviewed in Sect. 5.
**2) Case study and model description**
**2.1) Case study overview**
The dust event simulated for this study occurred during August 2-5, 2016 across the Arabian Peninsula, and
originated from a combination of synoptic and mesoscale dust sources. An example of the meteorological setup and
dust fields for this case study can be found in Fig. 1-2. For this event, the high summertime temperatures in the
desert of the Arabian Peninsula produce a thermal low couplet at the surface, with one low centered over Iraq and
the other over the Rub' al Khali desert in Saudi Arabia (Fig. 1.c). The local low-pressure couplet leads to cyclonic
surface winds between these two areas (Fig. 1.e), comprised of northerly flow from Iraq into Saudi Arabia, with
retuning southerly flow from Oman over the Persian Gulf and into Kuwait, and is a major non-convective
contributor to the dust budget for this case study (Fig. 1.f). In addition to these large-scale flow patterns, a daytime
sea breeze brings moist, maritime air from the coast of Yemen and Oman inland into the otherwise arid Saudi
Arabian basin (Fig. 1.e and 1.d). This moisture gradient is also evident in the skew-t diagrams, which represent an
inland radiosonde release site at Riyadh (Fig. 2.a), and a site closer to the coast in Abha (Fig. 2.b), both located in
Saudi Arabia. There is a stark difference in low-level moisture between the two sites, although both display a
subsidence inversion aloft between 500 and 600 hPa. Furthermore, nocturnal low-level jets form along the Zagros
mountains in Iran and Iraq, and the Red sea, both of which have been studied previously in the literature
(Giannakopoulou and Toumi, 2011; Kalenderski and Stenchikov, 2016).
Due to the region's inherent moisture constraints, convection is limited spatially to the coastal regions of the
Arabian Peninsula, as is most summertime convective and non-convective precipitation in this region (e.g. Shwehdi,
2005; Almazroui, 2011; Hasanean and Almazroui, 2015; Babu et al., 2016). Moist convective cells develop along a
low-level convergence line between the northerly basin flow and sea breeze front (Fig. 1.g and 1.h) aided by
elevated terrain in Yemen and Oman (Fig. 1.a). This convective setup along the southern portion of the Arabian
Peninsula is a feature evident in each day of this case study, initializing diurnally in the local late afternoon and early
evening, and thereby providing three days of data for analysis, with the height of convective activity occurring on
August 3rd. Individual convective cells form along the convergence line, a typical Middle Eastern characteristic
(Dayan et al., 2001), but do not organize further, owing to a lack of upper-level synoptic support and insufficient
moisture in the interior of the peninsula. Nevertheless, the convective line does produce outflow boundaries, which
loft dust from the surface and are the main convective dust source for this case study. More information on the
meteorological setup of this case study, including comparisons with aerosol optical depth (AOD) observations can
be found in Saleeby et al. (2019).



**2.2) WRF-Chem model description and physics**
To investigate the Arabian Peninsula case study, WRF-Chem version 3.9.1.1 is used to simulate the dust outbreak
meteorology and aerosol fields. WRF-Chem is an online numerical chemical transport model that allows for
interactive aerosol processes, including feedbacks between the meteorology, aerosol, and radiation. The model is
coupled to the Goddard Chemistry Aerosol Radiation and Transport (GOCART) module (Ginoux et al., 2001),
which will be described in more detail in Sect. 2.3.
The meteorological and sea surface temperature initial and lateral boundary conditions are sourced from the 0.25
degree, 6-hourly Global Data Assimilation System Final Analysis (GDAS-FNL). No chemistry or aerosol initial /
lateral boundary conditions are used. Rather, the aerosol fields are initialized with zero concentrations and are
allowed to evolve naturally from the model meteorology, aerosol, surface and radiation processes. The model is run
from 00:00:00 UTC on 02-Aug-2016 to 00:00:00 UTC on 05-Aug-2016 producing output at 30-minute intervals.
The following model parameterizations were employed and kept constant throughout the simulations: Morrison
double-moment microphysics (Morrison et al., 2005; 2009), RRTMG longwave scheme (Iacano et al., 2008),
Goddard shortwave radiation scheme (Chou and Suarez, 1999), the Noah Land Surface Model with
multiparameterization options (Niu et al., 2011; Yang et al., 2011), and the MYNN level 3 boundary layer
parameterization (Nakanishi and Niino, 2006; 2009). The convective parameterizations and horizontal resolutions
tested will be discussed in Sect. 2.4. A summary of the physics options utilized can be found in Table 1.
**2.3) GOCART dust emissions and dust uplift potential**
WRF-Chem is coupled to the GOCART dust module, which parameterizes the emission of dry mineral dust mass
from the surface to the atmosphere for 5 effective radii bins [0.5, 1.4, 2.4, 4.5, and 8.0 μm] based on Eq. (1):
$$F_p = CSs_p\, U^2(U - U_t)\ \ if\ U > U_t \tag{1}$$
In Eq. (1), $F_p$ is the dust flux from the surface [kg m$^{-2}$ s$^{-1}$] for each of the radii bins ($p$), $S$ represents the wind erosion
scaling factor [0 to 1] established by the Ginoux et al. (2004) soil erodibility map, $s_p$ is the fraction of each size class
within the soil [0 to 1] based on the silt and clay fraction of the soil type, $U$ is the 10 m wind speed [m s$^{-1}$], and $U_t$ is
the threshold velocity of wind erosion [m s$^{-1}$]. $C$ is a tuning constant (set here to a default 1 kg s$^2$ m$^{-5}$), which can be
set by the user to increase or decrease the total dust flux based on regional observations (e.g. Zhao et al., 2010;
Kalenderski et al., 2013; Dipu et al., 2013). If the wind speed is less than the threshold velocity, no dust will loft
from the surface. Most of the terms in Eq. (1) are time invariant ($C,S,s_p$), except for the wind speed ($U$) and wind
erosion threshold ($U_t$). $U_t$ is a function of soil wetness, and is calculated with the relationship found in Eq. (2):
$$U_t = \begin{cases} 6.5\sqrt{\dfrac{\rho_p - \rho_a}{\rho_a} g D_p}(1.2 + log_{10} w_{soil}) & if\ w_{soil} < 0.5 \\ \infty & if\ w_{soil} \geq 0.5 \end{cases} \tag{2}$$
For Eq. (2), $\rho_p$ is the dust particle density [kg m$^{-3}$], $\rho_a$ is the density of air [kg m$^{-3}$], $g$ is gravitational acceleration [m
s$^{-2}$], and $w_{soil}$ is the soil wetness fraction [0 to 1]. Similar to Eq. (1), Eq. (2) includes a threshold, whereby above a





soil wetness of 0.5, no dust will be emitted. If the threshold criteria are met and dust lofts from the surface, it is then
transported based on the simulated meteorological fields from WRF and is removed from the atmosphere via
gravitational settling and wet deposition. Here, wet deposition is included as a scavenging mechanism to provide a
more realistic picture of the convective transport process. Aerosol radiation interactions in the shortwave and
longwave (Barnard et al., 2010) are included in the simulations to understand the implications that lofted dust has on
the energy budget of the case study and are discussed in Sect. 3.3.
Before dust can amass in and influence the atmosphere, it must first be emitted from the surface. Because of the
threshold values included in the GOCART dust parameterization equations (Eq. 1-2), it is important to understand
how often the modeled near-surface wind speeds exceed the wind threshold value. A parameter useful in describing
the influence of the wind on dust emissions is Dust Uplift Potential (DUP), proposed by Marsham et al. (2011) and
based on Marticorena and Bergametti (1995). The DUP parameter is an offline approximation for the relative
amount of dust expected to loft from the surface. DUP is a convenient way to perform first order sensitivity tests on
the meteorology without having to re-run the model, and provides a framework for deconvolving the variables in Eq.
(1-2). Here, we have adapted the DUP parameter from Marsham et al. (2011) (Eq. 4) into three variations (Eq. 3-5),
which allows researchers to vary the complexity of the analysis by including more, or fewer degrees of freedom.
$$DUP(U) = U^3 \left(1 + \frac{A}{U}\right)\left(1 - \frac{A^2}{U^2}\right) \qquad (3)$$
$$DUP(U, U_t) = U^3 \left(1 + \frac{U_t}{U}\right)\left(1 - \frac{U_t^2}{U^2}\right) \qquad (4)$$
$$DUP(U, U_t, S) = SU^3 \left(1 + \frac{U_t}{U}\right)\left(1 - \frac{U_t^2}{U^2}\right) \qquad (5)$$
In Eq. (3), $U_t$ is set to a constant wind speed, $A$, thereby making DUP a function of only the near-surface wind
speed; for the purpose of this paper $U_t$ is set to 5 m s$^{-1}$, but has been tested elsewhere across the range of 5-10 m s$^{-1}$
(e.g. Marsham et al., 2011; Cowie et al., 2015; Pantillon et al., 2015). Eq. (4) is slightly more intricate in that it
considers the model evolution of $U_t$ due to changing soil wetness from precipitation and land-surface processes,
calculated by Eq. (2). Lastly, Eq. (5) builds on Eq. (4) by including the soil erodibility scaling factor ($S$), which
recognizes that the $U$ and $U_t$ relationship is valid only if it occurs over potential dust source regions. Since $U$, $U_t$, and
S are entangled in the GOCART dust parametrization found in Eq. (1-2), the seemingly minor variations between
the DUP parameters in Eq. (3-5) are crucial for isolating which processes, or combination of processes, are sensitive
to the horizontal resolution of the model, and hence to the analysis performed here.
**2.4) Domain, nesting, and cumulus parameterizations**
Several horizontal model grid-spacings (45 km, 15 km, and 3 km) of the Arabian Peninsula domain (Fig. 3) are
tested to identify the sensitivity of modeled dust concentrations to the model's horizontal resolution. For the two
coarsest simulations (45 km and 15 km), cumulus parameterizations are employed to represent shallow and deep
convection. The 45 km simulation was run with only the Betts–Miller–Janjic (BMJ) cumulus parameterization
(Janjic, 1994), while five different cumulus parameterizations were tested for the 15 km simulations, including the



BMJ, Kain–Fritsch (KF) (Kain, 2004), Grell 3D Ensemble (GD) (Grell, 1993; Grell et al., 2002), Tiedtke (TD)
(Tiedtke, 1989; Zhang et al., 2011), and Simplified Arakawa–Schubert (AS) (Arakawa and Schubert, 1974; Han and
Pan, 2011) schemes, which will determine the sensitivity of dust lofting to different cumulus parameterizations. The
finest resolution simulation (3 km) is run at convection-permitting scales and hence no cumulus parameterizations
were invoked. The 3 km simulation is a one-way nest initialized from the 15 km BMJ simulation which serves as its
parent lateral boundary conditions. A summary of the model domains is also found in Fig. 3.
The cumulus parameterizations tested in this study for the 15 km simulations vary in their methods for triggering
and then characterizing convective processes at the sub-grid scale level. BMJ is a moisture and temperature
adjustment scheme that acts to restore the pre-convective unstable thermodynamic profile to a post-convective stable
and well-mixed reference profile, while the other cumulus parameterizations (KF, GD, TD, AS) employ a mass-flux
approach to determine updraft and downdraft mass transport. Across the mass-flux parameterizations, GD is unique
in that it computes an ensemble of varying convective triggers and closure assumptions and then feeds the ensemble
mean back to the model. Furthermore, all five schemes represent shallow convection in addition to deep convection,
the mass-flux schemes include detrainment of water and ice at cloud top, and AS and TD are formulated to include
momentum transport in their calculations. These differences across parameterizations will result in varying updraft
and downdraft speeds and precipitation rates, which will have consequences for the vertical transport of airborne
dust, as well as the strength of convective outflow boundaries and therefore dust emission at the surface.
**2.5) Averaging and analysis methods**
Because the representation of convective processes varies across the simulations, the results will focus on composite
statistics from the three-day case study. The authors make no attempt to track and match individual convective
elements across simulations, as their triggering, timing, and development (or lack of development) will fluctuate
depending on the model resolution and cumulus parameterization, thus making a truly consistent analysis
problematic. Instead, this paper takes a step backward and aims to quantify in an average sense, how the choice of
horizontal resolution and parameterized convection affects dust concentrations in the WRF-Chem model across the
Arabian Peninsula. The analyses and averages are processed within the yellow box shown in Fig. 3, disregarding all
other grid points outside the Arabian Peninsula study area. Analyses that are averaged in time are only averaged
over the last two days of the simulation (00:00:00 UTC on 03-Aug-2016 to 00:00:00 UTC on 05-Aug-2016) to
account for model spin up in the first 24 hours. All results are summed over the five dust bins in the GOCART
model rather than being treated separately. Lastly, the results from the five 15 km simulations are averaged together
to produce a mean 15 km resolution response, and is presented, along with the maximum and minimum spread
across these simulations for reference.




**3) Results**
**3.1) Temporal evolution**
**3.1.1) Dust uplift potential**
The first process of interest in determining the sensitivity of modeled dust concentrations to horizontal resolution in
WRF-Chem is the amount of dust lofted from the surface to the atmosphere. Fig. 4 depicts the average DUP for the
simulations at each 30-minute output, using Eq. (3-5) to separate out the importance of the different mechanisms
regulating dust emissions.
Regardless of which DUP parameter is used, almost all of the simulations capture the bimodal daily maximum in
dust emissions in the local mid-morning (6 UTC) and late afternoon (13 UTC) due to the mixing of the NLLJ to the
surface and convective outflow boundaries, respectively. The only resolution where the bimodality is absent is the
45 km simulation, which captures the NLLJ mechanism, but misses the second convective activity maximum. The
coarsest simulation overestimates the NLLJ wind speeds, which subsequently inhibits convection later in the day.
Because of this, the 45 km simulation has the highest DUP(U) (Fig. 4.a) based only on wind speed (Eq. 3), a result
similar to the Heinhold et al. (2013) and Marsham et al. (2011) studies over the Sahara.
However, when taking the calculated threshold wind velocity into account (Eq. 4), the explicit simulation (3 km)
displays the strongest $DUP(U,U_t)$ at the local late afternoon convective maximum (Fig. 4.c). For this to be the case
compared to the DUP(U) parameter, the 3 km simulation must have a lower threshold wind velocity (Fig. 5.a) than
the simulations with parameterized convection. Since the threshold wind velocity is proportional to soil wetness (Eq.
2), this implies that the explicit simulation will on average have drier soil, or more grid points below the soil wetness
threshold than the parameterized simulations. The effects of drier soil are indeed evident in the surface fluxes with
the Bowen ratio of sensible to latent heat fluxes in Fig. 5.c. When the Bowen ratio is above one, more of the surface
heat exchange with the atmosphere is in the form of sensible heat flux, rather than latent heat flux. Dry soils are
characterized by low values of latent heat flux, and therefore exhibit higher Bowen ratios. The 3 km simulation
exhibits a higher Bowen ratio on August 3[rd] and 4[th], indicating that the soil is on average drier in the explicit
simulation. This result implies that disparities in land surface properties across the varying model grid resolutions
are important for modulating dust emissions, both from the perspective of explicit versus parameterized convection
and associated precipitation, as well as latent and sensible heat fluxes.
Adding on to the complexity of the DUP parameter, when the location of dust sources is considered in the
$DUP(U,U_t,S)$ calculations (Eq. 5), some of variability between the local NLLJ and convection maxima is lost in the
3 km simulation (Fig. 4.e) on August 3[rd]. Also, including the scaling factor reduces the magnitude of the DUP
parameter to roughly 10% of the initial values for DUP(U) and $DUP(U,U_t)$. Incorporating the dust source function in
DUP works not only as a scaling factor for the magnitude of potential dust emissions, but also impacts the relative
importance of dust production mechanisms (NLLJ versus convection). This shift is a consequence of the location in
which these processes occur. For instance, the reduction in the 3 km convective maximum on August 3[rd] between





DUP(U,U$_t$) and DUP(U,U$_t$S) signifies that convection is occurring in locations that are not active dust source
regions. Without information on the dust source regions, this process would be assigned an unrealistic dominance
over the NLLJ mechanism in terms of DUP.
All simulations are similar for the first 24 spin-up hours until the processes begin to diverge on August 3$^{th}$, where
the explicit simulation produces the maximum DUP(U,U$_t$S) both during the local daytime and nighttime hours. On
the final day of the case study (August 4$^{th}$), the explicit simulation has the lowest DUP(U,U$_t$S), with the NLLJ
maximum dominating over the convective maximum in both the 3 km and the 15 km mean, due to reduced
convective activity in the fine resolution simulations. Examining the percent difference in DUP between the coarse
and fine simulations (Fig. 4.b,d,f), the average percent difference between the 3 km and 15 km simulations is at a
minimum when only wind speed is considered, and increases as the degrees of freedom in DUP increases. For the
DUP(U,U$_t$S) case, the average percent difference is between 10-65% lower in the 15 km simulations than the
explicit simulation, with a maximum difference of 85% and a spread across parameterizations of 20%. This implies
that the explicit WRF-Chem simulation has the potential to loft up to 85% more dust than those with parameterized
convection.
**3.1.2) Integrated dust mass**
The differences in DUP(U,U$_t$S), or dust flux from the surface to the atmosphere, specifically the enhanced values
for the explicit simulation on August 3$^{rd}$, will lead to more dust lofting than in the coarse simulations. To see how
differences in the dust emissions translate into differences in airborne concentrations of dust, Fig. 6 demonstrates the
temporal evolution of the average integrated dust mass throughout the vertical column. Here, the explicit simulation
records upwards of 150% more integrated dust mass compared to the coarse resolution simulations. Across the
coarse simulations, the 45 km and 15 km runs have similar integrated dust magnitudes, despite the temporal
differences in DUP(U,U$_t$S). This is due to the overestimation of the NLLJ in the 45 km simulations being offset by
the enhanced convective dust lofting in the 15 km simulations.
The discrepancy in the diurnal maxima across horizontal resolutions is similar to the results of Marsham et al.
(2011) and Heinhold et al. (2013). Yet, the results here differ in that both of these previous studies found a stronger
NLLJ response in 12 km simulations with convective parameterizations than was found here in the 15 km
parameterized ensemble. In contrast to the findings of Marsham et al. (2011) and Heinhold et al. (2013), dust
emissions and airborne dust mass increases in the WRF-Chem simulations as resolution increases, which is in closer
agreement to the studies of Reinfried et al. (2009) and Bouet et al. (2012).
The temporal trends in integrated dust mass lag behind those observed in the DUP plots in Fig. 4. Particularly at
timesteps where DUP decreases, the change in integrated dust mass follows several hours later. The time series of
gravitational settling rates at the surface (Fig. 5.b) also lags behind the DUP trends, which implies that the removal
mechanisms for dust take time to act on the airborne particles once they are emitted. The rates of gravitational
settling are higher in the explicit simulation compared to the coarse simulations, yet Fig. 6.a suggests that this is not





enough to offset the higher dust emissions, or the integrated dust quantities would be similar across all the
simulations. The fact that the integrated dust values are higher in the 3 km simulation, despite higher rates of
gravitational settling, implies there must be a mechanism that acts to keep dust suspended longer in the explicit
simulations than in those with parameterized convection. There are clearly more processes occurring above the
surface to influence the integrated dust quantities than just a simple surface emission to surface deposition ratio.
This will be further deconstructed by examining vertical profiles in the following section.
**3.2) Vertical characteristics**
**3.2.1) Vertical dust and velocity profiles**
Moving away from vertically integrated quantities to a time and domain averaged vertical snapshot of dust (Fig.
7.a), the vertical dust profile follows a generally exponentially decreasing function and tapers off to low dust
concentrations in the range of 5-6 km above ground level (AGL). A widespread subsidence inversion is present near
6 km throughout the case study time period over the inner basin of the Arabian Peninsula (Fig 2), acting as a cap on
vertical motions and dust transport. Because dust concentrations do not vary much above this height, the plots in
Fig. 7 have been truncated at 9 km. There is a higher concentration of dust at every level in the explicit simulation
compared to that in the coarse simulations. Examining the percent difference plot between the explicit and other
simulations in Fig. 7.b, there is a difference of approximately 80% at the surface, which increases upwards to
~180% at 6 km. Above this level, the percent difference between the explicit and coarse simulations changes sign,
but the overall concentration is extremely low, and as such, the authors make no attempt to assign meaning to the
differences above 6 km.
For dust to reach higher levels in the atmosphere, it must have undergone vertical transport to move it aloft from its
initial source region at the surface. Several mechanisms could be responsible for vertical dust transport in the
Arabian Peninsula, including flow over terrain, daytime mixing (dry convection), and lastly, moist convective
updrafts, whose representation (explicit versus parameterized) is a defining difference between the horizontal
resolutions tested in this paper. Investigating the effect that increasing resolution has on updraft and downdraft
strength can be found in Fig. 8, which represents the mean of all vertical velocities above or below 0 m s$^{-1}$, including
points that are not vertically continuous. As resolution increases, the average range in vertical velocity also
increases. The simulations with parametrized convection have lower mean updraft / downdraft speeds than the
explicit simulation, on the order ~75% weaker near the surface for the 15 km runs and ~110% weaker for the 45 km
run. Irrespective of resolution, the mean updraft speeds in the WRF-Chem simulations are slightly higher than the
downdraft speeds near the surface, while at the surface mean downdraft speeds are higher than updraft speeds, a
consideration that will be discussed further in Sect. 3.2.2.
**3.2.2) Vertical dust flux**
The implications for dust transport based on vertical velocities is convoluted. As noted in Jung et al. (2005),
convective updrafts will lift aerosol particles upward into the free atmosphere, while downdrafts simultaneously





limit the maximum vertical extent of these particles. However, the convective transport simulations in Jung et al.
(2005) demonstrate that these opposing processes do not act as equal opposites in time, magnitude, and space. This
canon holds true for the Arabian Peninsula simulations as well. Fig. 9 contains Contoured Frequency by Altitude
Diagrams (CFADs) of vertical velocity (Yuter and Houze, 1995) normalized by the total number of grid points in
each simulation. The normalization is performed to remove an artificial larger frequency in the higher resolution
simulations that arises because there are more grid spaces available to count. Because no vertical velocity threshold
is imposed, a majority of points straddle zero. To highlight variability away from the zero line, the CFAD contours
are plotted on a log scale.
Similar to the mean plots in Fig. 8, as resolution increases, so does the variability in updraft and downdraft speeds.
There is a striking difference between the spread in vertical velocities at all altitudes across the 45 km, 15 km mean,
and 3 km simulations in Fig. 9. In the 45 km run, most of the velocities straddle +/- 1-2 m s$^{-1}$, whereas the explicit
simulation ranges from -10 to 30 m s$^{-1}$. Not only is the range larger, but the normalized frequency is greater in the
fine resolution simulation as well. The inference here is that stronger updrafts will transport dust higher in the
atmosphere, and that stronger updrafts are observed more frequently in the explicit simulation, thereby enhancing
the integrated dust transport.
Combining the information on the vertical distribution of dust and updraft / downdraft speeds, it is possible to
calculate a domain averaged dust flux profile (Fig. 8). Again, the magnitude of the dust flux upwards and
downwards from the surface through 6 km AGL is higher in the explicit simulation compared to the parametrized
simulations. Moreover, the mean near-surface upwards dust flux is stronger than that for the downward dust flux,
which coincides with the mean updraft speeds being slightly higher than the mean downdraft speeds at these same
vertical levels (Fig. 8). This relationship also holds in the dust flux CFADs (Fig. 9), in which the upward and
downward flux of dust has more variability in the 3 km simulation, and stronger vertical dust fluxes are more
frequent.
Similarly, there is more dust transport evident at higher vertical levels in the explicit simulation, which has
implications for the residence time of the dust particles. As dust is transported higher in the atmosphere, absent any
sort of external motion or coagulation outside of gravitational settling, the atmospheric lifetime of the particles will
increase. Figure 10 shows the theoretical terminal velocity of dust particles in WRF-Chem using the Stokes settling
velocity with slip correction for pressure dependence (Fig. 10.a) and their lifetime based on different starting heights
in the atmosphere (Fig. 10.b), which increases exponentially away from the surface. As such, dust in the explicit
simulation will take longer to settle out, leading to the higher observed integrated dust values (Fig. 5) compared to
the parameterized simulations. Looking at the distribution of downdrafts in the vertical velocity CFADs (Fig. 9),
there is a clear bimodal signal aloft in both the explicit and 15 km simulations, being representative of two distinct
subsidence layers, which act as a cap on vertical transport. The local minimum occurs around 6 km, which could
explain why dust fluxes also taper off at this level.



At the surface, higher dust flux values are found in association with the downdrafts, producing a pronounced
skewness towards high, yet infrequent values of strong negative dust flux towards the ground (Fig. 9). It is
hypothesized that this skewness is a consequence of the dissimilar background dust conditions in the vicinity of
near-surface downdrafts and updrafts, similar to the results found in Siegel and van den Heever (2012), which
studied the ingestion of dust by a supercell storm. Updrafts originate in relatively clear air, and will consume
background dust and transport it upwards. However, downdrafts occur through the cold pool, and hence their source
is, at least partially, within the dusty cold pool. As such, downdrafts will have access to more dust and thus transport
more of it in the downward direction. This skewness warrants further research, preferably from an idealized
perspective, to better understand the relationship between storm dynamics, dust emissions, and transport.
In all, the increased vertical and integrated dust concentrations in the 3 km run are a product of several processes
working together. Compared to the simulations with parameterized convection, the 3 km run has enhanced potential
for dust uplift due to stronger resolved downdrafts and lower wind velocity thresholds, higher vertical transport due
to more frequent, stronger updrafts, and a lengthier theoretical residence time once being lofted to higher levels.
**3.3) Impacts on radiation**
Beyond the first-order sensitivity of model resolution to dust emissions and concentrations for the Arabian Peninsula
case study, there are higher-order effects that disseminate from changing dust concentrations. One example being
the modification of atmospheric heating / cooling rates and the radiation budget due to dust absorption and scattering
(see Sect. 1). The domain and time averaged shortwave (SW), longwave (LW), and net dust heating / cooling rates
are found in Fig. 11. The average dust heating and cooling rates were calculated over the last 48 hours of the
simulation as a difference between the radiation tendency with dust aerosols and without. Ostensibly, since dust
concentrations increase in the model as resolution increases, so does the magnitude of the radiative effects. There is
a stronger SW cooling and LW heating effect in the 3 km simulation, and this trend follows the vertical distribution
of dust from Fig. 7, again tapering off near 5-6 km AGL.
Most interestingly, however, is the difference in the net aerosol heating rate. In the lowest layer (<1.5 km), there is a
sign change between the fine and coarse simulations. The SW effect in the explicit simulation is strong enough to
elicit a net cooling effect in this near-surface layer. Conversely, the LW aerosol heating effect dominates in the
coarse simulations, resulting in a net warming effect. The difference between warming and cooling can have
cascading effects on the thermodynamic profile, static stability, and future convective development, which in turn
impacts the relative importance between convection and the NLLJ discussed earlier. The sensitivity of dust
concentrations to horizontal model resolution is important to understand in its own right, but furthermore, this
sensitivity leads to higher-order changes in model predictions. If NWP models or GCMs are going to incorporate
dust radiative effects, concentrations need to be highly constrained, not only to accurately capture the magnitude, but
the sign of the response as well.



**4) Discussion and recommendations**
For this Arabian Peninsula event, horizontal resolution in the WRF-Chem model has a considerable effect on the
dust budget of the region. Because aerosol prediction models and GCMs still employ cumulus parameterizations, it
is important to discuss the uncertainties unearthed in this paper, as well as recommendations for past and future
forecasts and research that will be generated prior to our ability to consistently run these models at convection-
permitting resolutions.
In an average sense, there will be higher dust concentrations produced in explicit convection simulations compared
to those with parameterized convection. The major point here is that the uncertainty in dust concentrations for
simulations using different cumulus parameterizations (15 km ensemble), or using different horizontal resolutions
with the same cumulus parameterizations (45 km versus 15 km) is small relative to the differences between the use
of parameterized versus explicit convection. *Most of the uncertainty in the model's predicted dust concentrations*
*comes from the choice to either parameterize or explicitly resolve convection.*
The results of this research do not stand alone in the literature focused on the impact of horizontal model resolution
on dust emissions, and there are several similarities and differences to note when comparing this paper to previous
studies. Firstly, concerning the diurnal variation in dust emissions, we find a similar response in the NLLJ
mechanism to that of Heinhold et al. (2013) and Marsham et al. (2011), whereby the coarsest simulations
overestimate the early morning windspeeds caused by the mixing of the jet to the surface and fail to capture the late
afternoon / early evening convective dust lofting mechanism. In these previous studies, the explicit simulation
reduces the importance of the NLLJ and enhances the convective maximum, but still retains the NLLJ as the
dominant process for dust uplift. Overall, Heinhold et al. (2013) and Marsham et al. (2011) found a net reduction in
dust uplift with explicit convection. While the NLLJ mechanism is found to be similar here, the analysis reveals an
opposite response in WRF-Chem for the Arabian Peninsula, in which the convective maximum dominates, but the
NLLJ is still an important mechanism, which thereby leads to more, rather than less dust in the explicit simulations.
The net increase in dust concentrations in WRF-Chem is similar to the findings of Reinfried et al. (2009), although
Reinfried et al. (2009) focused mainly on haboobs, which may point to convection being the source of agreement
rather than the balance between the NLLJ and convection. At this point, we cannot determine whether the
discrepancies between our results and previous literature comes from regional or case study differences in the
importance of these mechanisms to the dust budget, differences in the models' representation of these processes, or a
combination of the two. In all, more work needs to be done to investigate the relationship between the NLLJ and
subsequent late afternoon convection in dust producing regions, and the representation of this in numerical models.
From an integrated viewpoint, for the Arabian Peninsula region it is possible to rudimentarily tune the dust
concentrations of the coarse simulations to that of the explicit simulation by multiplying by an average constant
derived from the dust difference plots in Fig. 6-7, which would be on the order of ~2. This is an offline solution,
which would aid in enhancing the accuracy of a first-order forecast of integrated or surface dust, and/or AOD.
Nevertheless, attempting to use this tuning parameter online in the model (i.e. adjusting the tuning constant, *C,* in



Eq. 1) would not reconcile the differences from a dust flux standpoint. Even if more dust were to be emitted from
the surface, the parameterized simulations still lack the necessary variability in updrafts and downdrafts, especially
updraft strength, to transport the dust upwards and away from the surface, thus misrepresenting the atmospheric
lifetime of these particles in the process.
Moreover, tuning the dust concentrations will not change the effect horizontal resolution has on the soil
characteristics, particularly soil moisture, and hence on the a priori determined threshold wind speeds which are
important in calculating dust lofting in the first place (Fig. 4). If dust concentrations are inaccurately predicted in the
coarse simulations, or erroneously tuned, the higher-order online feedbacks will also be incorrect, such as
modifications to the radiative budget, and feedbacks to the thermodynamic profile, static stability and mesoscale
features, particularly those driven by differences in thermodynamic gradients, such as sea breezes and cold pool
propagation.
**5) Conclusions**
In this study, we have quantified the response sign and magnitude in modeled dust fields in the WRF-Chem regional
model to increasing horizontal resolution and the manner in which convection is represented for a summertime
Arabian Peninsula event. We have investigated the variability in dust concentrations and fluxes due to the choice of
convective parameterization, the representation of convection in the model (explicit versus parameterized), and the
effect these differences in dust concentrations have on aerosol heating rates. The case study was simulated at three
different horizontal resolutions (45 km, 15 km, and 3 km), with the two coarsest simulations run with cumulus
parameterizations, and the 3 km simulation run at convection-permitting resolution. To understand the uncertainty
across different parameterizations, five separate cumulus parameterizations were tested in an ensemble (BMJ, AS,
GD, TD, KF) at 15 km grid spacing.
The explicit convection simulation exhibited a stronger potential for dust uplift as a function of modeled wind speed,
wind threshold, and the location of dust sources. The wind threshold for dust lofting in the 3 km simulation was on
average, lower than that for the 15 km or 45 km. This is due to differences in grid resolution leading to changes in
the soil moisture, whereby the 3 km simulation displays lower soil wetness across the domain. Furthermore, a
distinct difference across simulations was identified in the representation of the bimodal daily maximum in dust
emissions in the local mid-morning (mixing of the NLLJ to the surface) and late afternoon (convective outflow
boundaries). Compared to the 3 km case, the 45 km simulation overestimates the contribution from the NLLJ and
underestimates the role of convection in dust emissions.
The 3 km simulation also produced higher integrated dust values at every timestep, as well as higher dust
concentrations at every vertical level in the lower troposphere (below 6 km AGL). The uncertainty in dust
concentrations for simulations using different cumulus parameterizations (15 km ensemble spread) is much smaller
than the difference between the parameterized and explicit convection cases. For the WRF-Chem Arabian Peninsula
simulations, the modeled dust fields were most sensitive to the choice of parametrizing or explicitly resolving



convective processes. The enhanced dust concentrations in the explicit case are the result of stronger downdrafts
lofting more dust from the surface, and stronger updrafts carrying dust to higher levels of the atmosphere, thereby
increasing the airborne lifetime of the dust particles. The difference in dust mass across the simulations leads to a
significant modification of the radiation budget, specifically the aerosol heating rate. The explicit simulation
revealed a greater shortwave and longwave effect, and for aerosol heating rates in the lowest levels, shortwave
cooling is stronger than longwave heating, leading to a net cooling effect. Conversely, the opposite radiative
response is present in the parameterized cases, resulting in a net warming effect, causing a change in sign in the
lowest levels compared to the explicit convection case.
There are a number of implications these results may have on forecasting and future studies. The dust concentrations
in the coarse simulations could be tuned offline to match those in the explicit simulation using the percentage
difference plots included in Fig. 5-6. This tuning would be on the order of ~2. However, because vertical transport is
essential to the vertical concentrations and lifetime of the particles, this tuning factor cannot be applied online. Even
if such a tuning were applied, this change will not accurately capture higher-order feedbacks to the meteorology,
thermodynamic environment and radiation budget of the Arabian Peninsula, or to the soil moisture wind threshold
velocities. Finally, this work also points to the need to better constrain dust concentrations in numerical models, and
further develop our understanding of the relationship between storm dynamics and dust processes.
**Author contributions**
Jennie Bukowski (JB) and Susan C. van den Heever (SvdH) designed the experiments. JB set up and performed the
WRF-Chem simulations and wrote the analysis code. Both JB and SvdH contributed to the analysis of the model
output. JB prepared the manuscript with contributions and edits from SvdH.
**Competing interests**
The authors declare that they have no conflict of interest.
**Acknowledgements**
This work was funded by an Office of Naval Research – Multidisciplinary University Research Initiative (ONR-
MURI) grant (# N00014-16-1-2040). Jennie Bukowski was partially supported by the Cooperative Institute for
Research in the Atmosphere (CIRA) Program of Research and Scholarly Excellence (PRSE) fellowship. The
simulation data are available upon request from the corresponding author, Jennie Bukowski. Initialization data for
the model was provided by: National Centers for Environmental Prediction, National Weather Service, NOAA, U.S.
Department of Commerce. 2000, updated daily. NCEP FNL Operational Model Global Tropospheric Analyses,
continuing from July 1999. Research Data Archive at the National Center for Atmospheric Research, Computational
and Informational Systems Laboratory. https://doi.org/10.5065/D6M043C6.



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





| WRF-Chem Version 3.9.1.1 | Parameterization / Model Option |
|---|---|
| Simulation Start | 02-Aug-2016-00:00:00 UTC |
| Simulation End | 05-Aug-2016-00:00:00 UTC |
| Domains | dx = dy = 45km / 15km / 3km |
| Nesting | One-way |
| Vertical Levels | 50 stretched |
| Initialization | GDAS-FNL Reanalysis |
| Aerosol Module / Erodible Grid Map | GOCART / Ginoux et al. (2004) |
| Microphysics | Morrison 2-Moment |
| Radiation | RRTMG Longwave & Goddard Shortwave |
| Land Surface | Noah-MP Land Surface Model |
| Cumulus Schemes (45 km and 15 km grids only) | Betts–Miller–Janjic (BMJ) |
| | Kain–Fritsch (KF) |
| | Grell 3D Ensemble (GD) |
| | Tiedtke Scheme (TD) |
| | Simplified Arakawa–Schubert (AS) |
| Boundary Layer / Surface Layer | MYNN Level 3 |

**Table 1: Summary of WRF-Chem model options utilized and the simulation setup.**

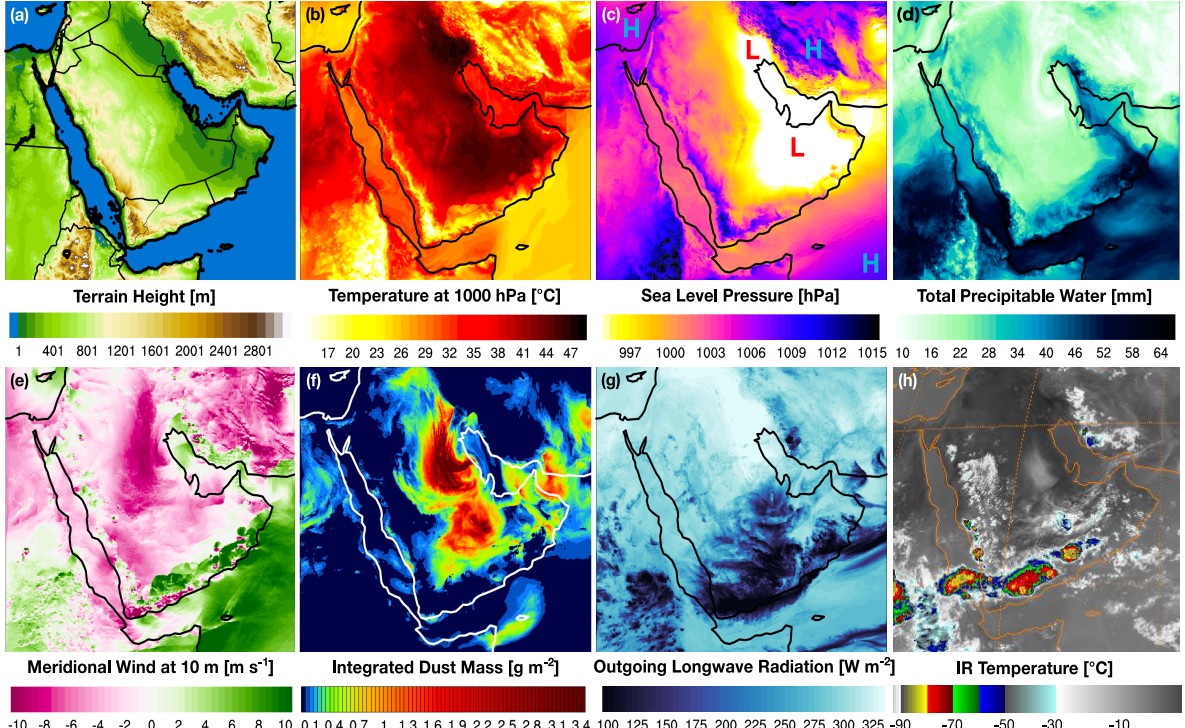

**Figure 1: Case study topography and meteorology for August 3, 2016 at 15:00 UTC: (a) terrain height and national**
**boundaries, (b) 1000 hPa Temperature, (c) sea level pressure, (d) total precipitable water, (e) meridional winds at 10 m**
**AGL, (f) integrated dust mass, (g) outgoing longwave radiation, and (h) IR temperature. Panel (h) is observed from**
**Meteosat-7 while panels (a-g) are snapshots from the 3 km WRF-Chem simulation**



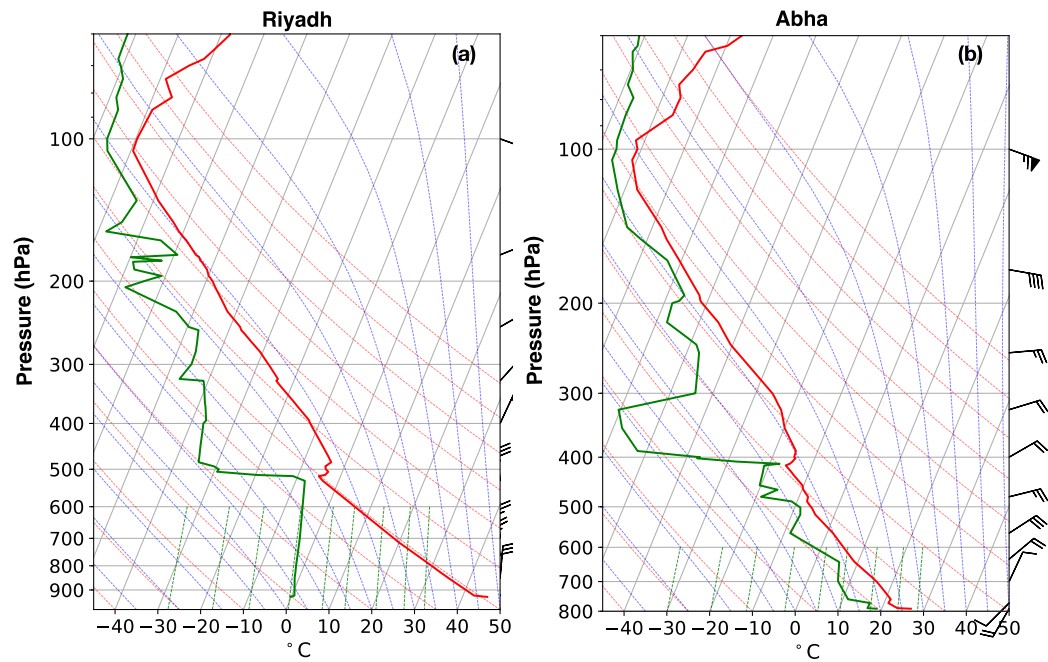


**Figure 2. Skew-T diagrams for two radiosonde release sites in Saudi Arabia on August 3, 2016 at 12:00 UTC for an inland location (a) and a location nearer to the coast (b).**





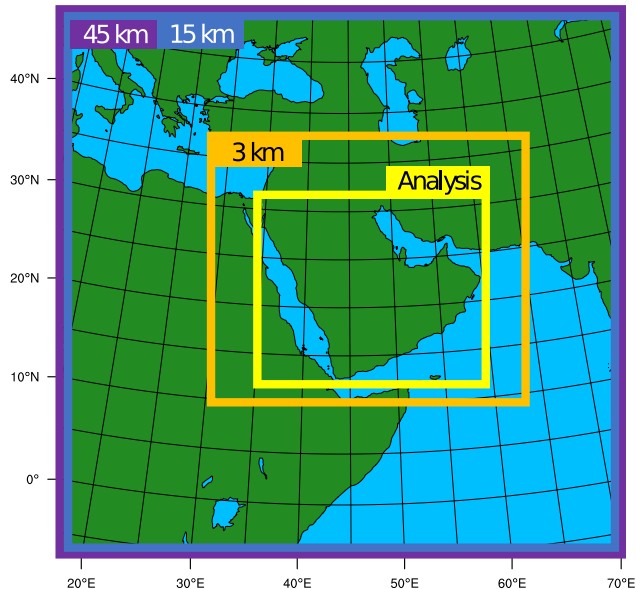


**Figure 3: Model domain setup and analysis region for the 45 km (purple) and 15 km (blue) independent simulations with**
**cumulus parameterizations, and the 3 km nested convection permitting simulation (orange). The averaging region for the**
**analysis is denoted in yellow.**





**Figure 4:** Left column: domain averaged dust uplift potential for (a) DUP(U), (c) DUP(U,$U_t$), and (e) DUP(U,$U_t$,S) for the 45 km (blue), 15 km mean (red), and 3 km (black) simulations with the maximum and minimum spread across the 15 km simulations indicated in light red shading. Note that in panel (e) there is a change in scale in the ordinate. Right column: percent difference between the 3 km convection-permitting simulation and the simulations employing cumulus parameterizations for the different DUP parameters.



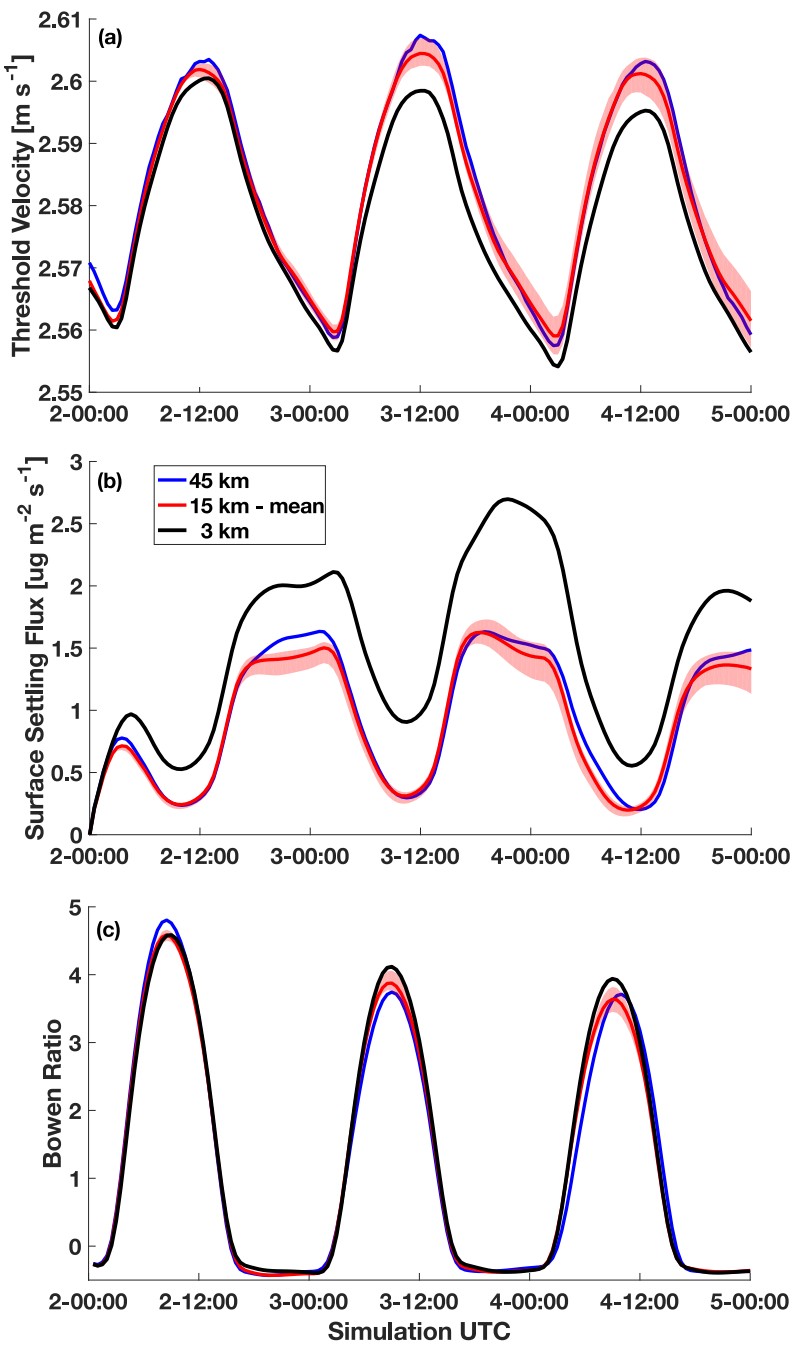


**Figure 5: Domain averaged (a) dust uplift threshold velocity, (b) dust surface settling flux, and (c) Bowen ratio of sensible**
**to latent heat flux. Colors and shading are the same as in Fig. 4.**





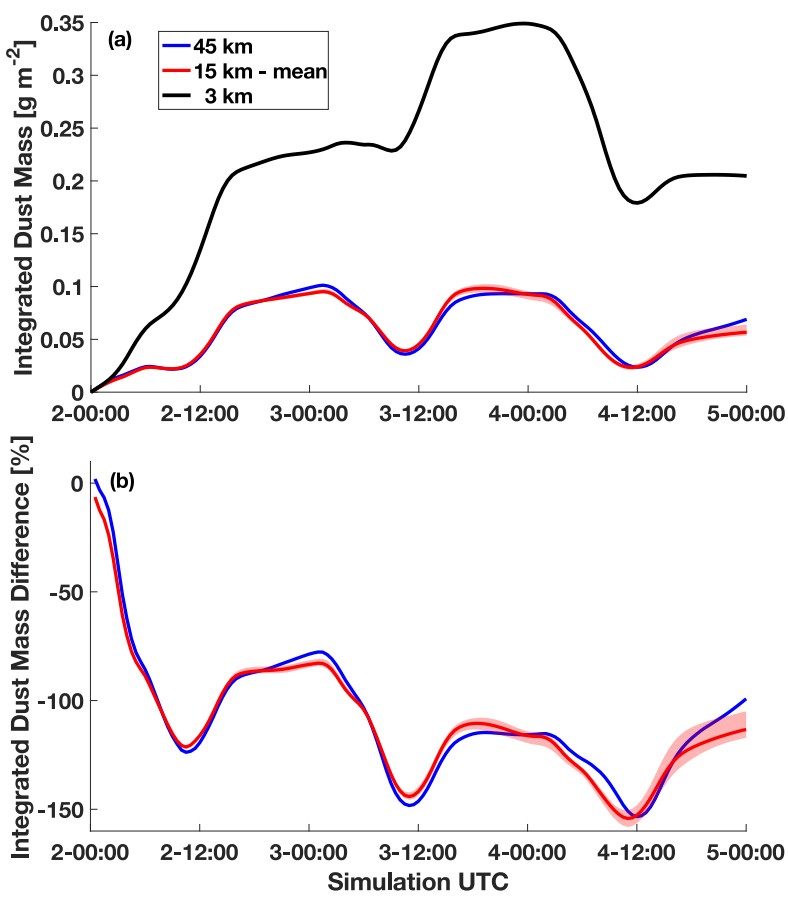


Figure 6: Domain averaged integrated dust mass. Colors and shading are identical to that in previous figures.



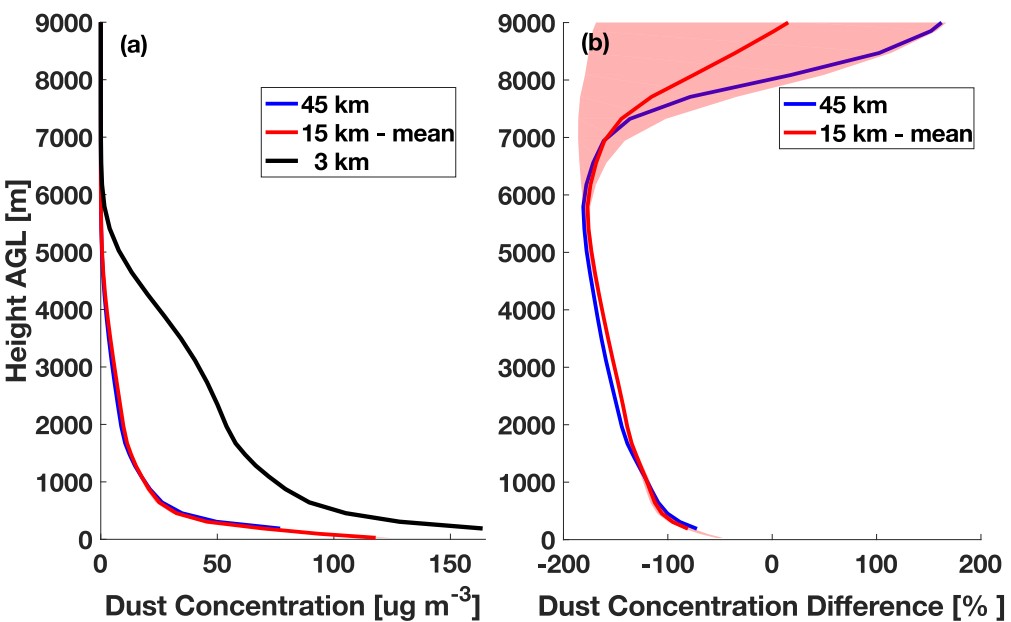


**Figure 7: Domain and time averaged vertical dust concentrations (a), with the (b) percent difference between the 3 km**
**convection-permitting simulation and the simulations employing cumulus parameterizations. Plots are truncated at 9 km**
**since the values above this height do not significantly vary from what is shown here. Colors and shading are identical to**
**that in previous figures.**





**Figure 8.** Left column: domain and time averaged vertical velocities (a), with the (b) percent difference between the 3 km convection-permitting simulation and the simulations employing cumulus parameterizations. All velocities above or below zero were considered. Colors and shading are identical to that in previous figures. Right column: same but for vertical dust mass flux. Note that in panels (c) and (d) the vertical axes are truncated at 9 km since the values above this height do not significantly vary from what is shown here.



862

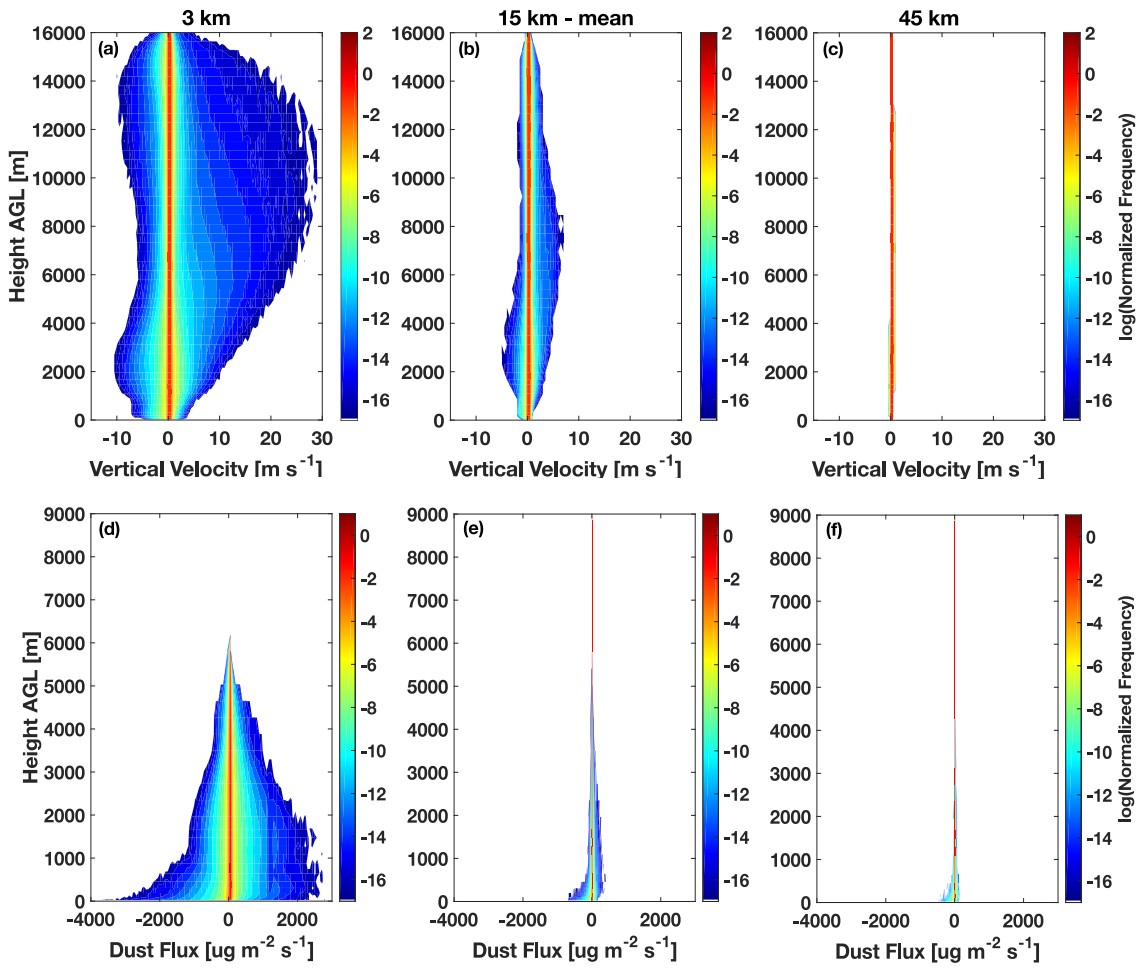

863

**Figure 9: Top row: Contoured Frequency by Altitude Diagrams (CFADs) for vertical velocity, normalized by the number of grid points in each respective simulation. The contours are computed on a log scale to highlight the variances away from zero. Bottom row: same but for vertical dust mass flux. Note that the panels in the bottom row are truncated at 9 km since the values above this height do not significantly vary from what is shown here.**





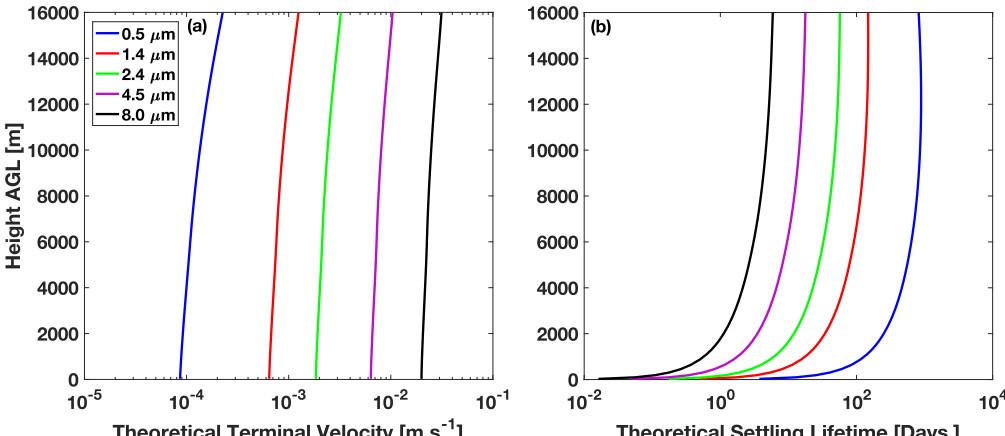

868

**Figure 10:** Theoretical terminal velocity of dust particles (a) based on Stokes settling velocity with slip correction for
pressure dependence for the 5 effective radii of dust particles in WRF-Chem. The calculations assume no vertical
motions, advection, deposition, coagulation, or condensation. (b) The lifetime of these theoretical dust particles based on
their height in the atmosphere.






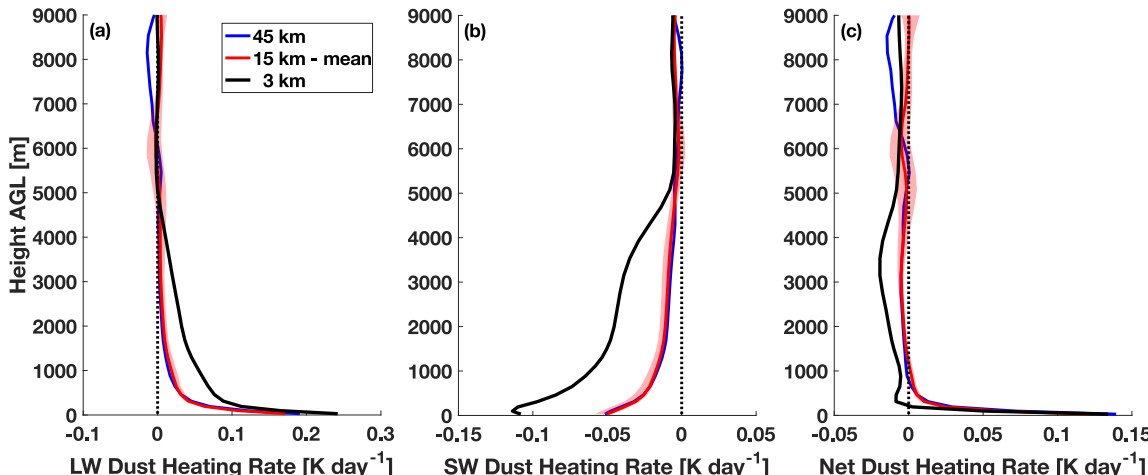


**Figure 11: Domain and time averaged longwave (a), shortwave (b), and net (c) dust heating rate profile for the 45 km**
**(blue), 15 km mean (red), and 3 km (black) simulations with the maximum and minimum spread across the 15 km**
**simulations indicated in light red shading. Plots are truncated at 9 km since the values above this height do not**
**significantly vary from what is shown here.**