# Peer review of "Convective distribution of dust over the Arabian Peninsula: the impact of model resolution"

_Atmospheric Chemistry and Physics, 2019_

## Referee Comment (RC1) · Anonymous Referee #1 · 10 May 2019

General comments:

The authors study the emissions and transport of mineral dust aerosol in the region of the Arabian Peninsula. In this region, along the coast, dust emissions may be huge, espcially during convective events. With a regional model (here WRFchem), emissions are primarily sensitive to the near-surface wind speed (in general the 10m wind speed, due to the parameterizations used). The authors made a sensitivity experiment by comapring dust emissions and concentrations, their impact on radiation, by using the same model but with different convection schemes. The 'reference' case is a simulation over the same region/period but with a resolution high enough to explicitly treat the

convection.

Detailed comments:

l.175: What is the meaning of 'coupled'? Probably only the use of the mineral dust emissions module (but not the transport, mizing, deposition etc.). Please better explain.

l.177 More details are needed about the schemes used. The paper is a sensitivity study about these schemes and they are not explained. In particular, the way to treat the aerosol for the indirect effects is completely different (the Grell scheme is aerosol aware compared to the others).

l.178: for a mineral dust study "no chemistry" why not. But no initial and boundary conditions for a simulation of 3 days, it is not possible to have realistic results.

l.182 "kept constant" meaning remain the same during the whole simulation?

l.214: The 'Dust Uplift Potential' is a calculation already done in a large majority of dust emissions schemes, by principle of the mechanism to evaluate. Unfortunately, it represents only a small part of the problem and is not really useful. It describes only the link between the friction velocity treshold (using the aeolian roughness length) and the current friction velocity. But other important parameters are not taken into account: the vegetation, the erodibility, the soil humidity, the recent precipitation etc. In addition, the fact to use a constant Ut is not realistic (eq.3): the aeolian roughness length is far to be constant over erodible region. It is the most important varying parameter in mineral dust emissions modelling. The use of three different kind of DUP has a large interest. The message is already contained in one. If the authors really want to use this criteria, only one is enough.

l.239: The reference simulation has an horizontal resolution of 3km to enable explicit convection calculation. This simulation has boundary conditions and this is a good point. But these boundary conditions are from the BMJ simulation, i.e one of the studied case. Thus, we can think that the reference case will be very influenced by this

case, no? To have a more realistic comparison between scheme, the 'reference' has to be done for each scheme and a first spread can be calculated between all 'high resolution' cases.

Figure 4: this figure clearly shows that the most important choice is the horizontal resolution and not the convection scheme.

l.263: for long-range transport, 24h of spin-up is not enough. For the time averaged results, it is only th elast two days. But for the time series, it is the 3 days? why this difference?

l.272: why not use directly the mineral dust emissions fluxes? Please explain this important point.

l.280: why the simulation with the coarsest resolution (and not simulation) overestimates the wind speed? Please explain (and I imagine it is the "10-m wind speed", please correct).

l.293: Yes, it is right. And obvious. Of course, a key point in modelling is to try to have a model not sensitive to the spatial resolution. And it seems it is the problem with WRF-chem. In WRF, the principle is to use, for each grid cell, the dominant soil type and landuse. Thus, by principle, the result is very sensitive to the resolution. Some other models are using subgrid scale variability and Weibull distribution for the 10-m wind speed, for example, to avoid this problem. Please see bibliography and replace WRF-chem in the context of all currently used regional dust models.

l.331: it is not sure that there is an interest to have a conclusion such as "resolution increases or decreases the mineral dust emission fluxes". In fact it depends on the studied area, the variability of the orography, aeolian roughness length, soil humidity, vegetation. And, of course, the way to well take into account or not all these processes and their variability.

l.335: I don't understand the discussion with "The rates of gravitational settling are
higher in the explicit simulation compared to the coarse simulations, yet Fig. 6.a suggests that this is not enough to offset the higher dust emissions, or the integrated dust quantities would be similar across all the simulations." The dry deposition is proportional to the concentrations, being a velocity applied to the concentrations. How is it possible to have 'enough' settling to 'offset' the higher dust emissions?

Figure 6: the fact to have difference sbetween resolution is understandable but a factor 2 has to be better explained. Mineral dust emissions mass maps for the common domain (the one with 3km horizontal resolution). The caption is not easy to understand: "Domain averaged integrated dust mass". Please correct with Spatially averaged, vertically integrated.

l.346: "the vertical dust profile follows a generally exponentially decreasing function" is it a conclusion of this study? or coming from a reference? These is no reason to have an exponential decrease in the troposphere. Many cases of thin but concentrated dust plumes transports are observed and modelled...

l.369: "The implications for dust transport based on vertical velocities is convoluted." This sentence is difficult to understand.

l.421: The impact on radiation, with potential heating and cooling, is a process needing more than 2 days of simulation to be significant.

l.428: there is a sign change. Could you explain why?

Conclusion of the comments:

The study suffers of several issues, mainly methodological.

1. There is no data used in this work: the simulations are compared between them but we have no idea of the realism of the simulations (there is only one reference for a comparison to Aeronet AOD in another paper, under discussion, and no guarantee this is exactly the same model set-up, and which one?). At least, the reference case (dx=3km) should be compared to available data (surface networks such as MIDAS,

AERONET, satellite, other data).

2. The studied case extended from 2 to 5 August 2016: there is no spin-up time, important when studying transport of aerosol such as mineral dust. Time series are presented for the three days, but some average are done only for the last two days, explaing that the first day is spin-up. But, viewing the domain size, the minimum spin-up time should be at least one week.

3. There is no boundary or initial conditions. These missing background values may have a large impact on the results, in particular knowing that the model couples the meteorology and the aerosol concentrations: direct and indirect aerosol effect may be long-term and it is required to have correct boundary conditions to have realistic effect of aerosol on meteorology. For the 'reference' domain, the boundary conditions are extracted from one of the studied case, biasing the results.

4. The convection schemes used are not explained. The paper is a sensitivity study about these schemes but there is no explanations about their real differences, how they take into account aerosol or not, thus no conclusion about why results may be different depending on the scheme.

5. The paper deals with the sensitivity to the model resolution. But since the schemes are not well implemented (no wind speed distribution, no subgrid scale variability), there is a large sensitivity but not for realistic and physical reasons: the differences are not due to the convection schemes in general but just to the fact that the problem of the resolution is not well designed in this model: it is not possible to describe a treshold problem (such as mineral dust emissions) without taken into account disstributions of input parameters. Results are linked to this model only and are not useful for other modellers.

In conclusion, I recommend 'rejection' to give a chance to the authors to really re-design the paper. It is obvious that all calculations have to be reprocess in order to have a minimum of confidence in the results. More important, the main scientific goal has

also to be revised: it is not possible to conclude for a sensitivity study, by running only 3 days over a large domain, with a coupled model and for mineral dust aerosol (long-range transported species, high SW impact), and without boundary/initial conditions.
* * *

---

## Referee Comment (RC2) · Anonymous Referee #2 · 6 Jun 2019

General comments

This paper aims to use the Weather Research and Forecasting model in its chemistry mode (WRF-Chem) to simulate the raising of dust for a specific case that occurred from 2nd to 5th August 2016 over the Arabian Peninsula. Simulations have been performed using a variety of grid spacings and convective parameterizations as well as explicitly representing convection in some cases.

Generally I think that this work is of a high standard and is well written. The reasoning and thoughts of the authors are clear and contextualised well in earlier literature. For this work to be published I would recommend only minor editorial changes.

[Figure]

I liked the of using different complexities of the DUP metric whereby different versions of DUP were generated and compared with one another. This method allows for each of the added levels of complexity to be assigned to a different physical property within the simulation and therefore the magnitude of the effects can be seen.

I know that it is common parlance in the community to refer to simulations that are run without the use of convective parameterizations as being "explicit" or that convection is is "explicitly represented". However, more recently there has been a shift towards the use of simulations of this type being referred to as "convection permitting". This difference is subtle but I think is a better descriptor of what the models are actually doing. The model grid-scales involved are not so fine as to explicitly resolve individual updraughts and downdraughts but are sufficiently high to permit the development of convective storms that approximate those that we might observe in reality. I feel that it would be better to replace descriptions of simulations currently described as explicit with convection permitting.

Did you consider running a 15 km simulation with the convective parameterisation switched off. I don't think that you should do this as the work is already of a high standard, but think that you might well be surprised at how small the difference is between a 15 km grid-spaced convection permitting simulation and a 3 km grid-spaced convection permitting simulation.

Specific comments

Ln 17 -20 You need to be clear that the updraughts that are transporting dust vertically are part of the general circulation (eddies) in the dry atmopshere. At first I thought you were specifically talking about storm updraughts (which I assume are less important in the simulation for vertical dust transport due to washout).

Ln 45-47 I think it would be wise to indicate that in reality ingestion of this type is impossible. What you are hoping for is that the initialisation data and the representation of dust are good enough for your purposes. It is perfectly possible that that is true for

this case study but that the same setup run for different case studies could provide different results due to the high dependency of models (even those that do not contain dust) on initial conditions.

Ln 47-49 Is it the global and regional nature of models that causes these differences or is it the grid-spacing or other model differences? Please be clear.

Ln 50 I would get rid of "accurately" here. Generally in models dust processes are fairly simplistic and highly parameterised and so the idea that dust processes are accurately represented is a fallacy.

Ln 53-59 This section needs rewording. The first sentence along with the word "Additionally" suggests that large-scale, synoptic-scale and meso-scale meteorology is separate from the phenomena listed below. Also why say large and synoptic scales? Instead I would suggest something like "Dust uplift events can be associated with meteorological processes across a broad range of scales. Synoptic scale uplift phenomena include monsoon troughs (Marsham et al, Beegum et al), Shamal winds (Yu et al.) and frontal systems (Beegum et al). While dyamical effects on smaller (meso) scales can raise dust through the production of convective outflow boundaries (haboobs; Miller et al.) and the morning mixing of nocturnal low level jet (NLLJ) momentum to the surface (Fiedler et al)."

Ln 60 What other drivers of dust emission are there? There are prerequisite conditions (dry, unvegetated surface etc.) but wind is the only driver of surface dust emission that I can think of.

Ln 73 Heinold used offline emission which I think is a relevant point to mention here as it significantly differs from your approach. Another paper that discusses the grid-scale effects on online model dust and convective representation of dust in West Africa would be Roberts et al. 2018 (doi.org/10.5194/acp-18-9025-2018).

Ln 82 One thing that you don't mention is that the thing that effects models the most

is not the grid scale, or the microphysics and in some cases not even the whether simulations are convection permitting or parameterized. It is the initialisation data. This is one of the findings in Schepanski et al. 2015 (doi.org/10.1002/qj.2453) in West Africa.

Ln 104-114 Roberts et al. 2016 (mentioned above) covers some of these points by using the Met Office Unified Model over West Africa. In the UM over summertime West Africa at least, the grid spacing does very little compared to representation of convection.

Section 2.1 I find the ordering here a little odd. I would normally expect the model description to precede the description of the conditions that caused the dust uplift. It feels a little like you are skipping backwards and forwards between results and methods. I advise moving your current section 2.1 to either the end of section 2 or the start of section 3.

Ln 144-145 I don't think that Figs 1 and 2 show this. The first shows a number of different fields (not dust) and I wouldn't describe Figure 1 as the meteorological setup either. Figure 2 is actually 2 profiles which doesn't match the description either.

Please be much clearer in you description. I cannot tell what you are referring to.

Ln 180-187 A very brief description of why these parameterizations were chosen would be welcome. For instance is this a replication of a setup used in a similar study? Is it similar to operational setups of WRF that are run for similarly arid regions? Or is there an individual reason for having chosen each of these options. Ln 289 You should say why the soil moisture is more likely to fall below the threshold in the convection permitting simulations. This is very likely associated with the different way in which rainfall in generated in parameterised and convection permitting simulations. Parameterized simulations have much more widespread light rainfall while convection permitting simulations have rainfall over much smaller areas but at much higher rates. The smaller areal coverage of rainfall in the convection permitting simulations is most probably the

cause of the soil moisture threshold not being exceeded as frequently.

Ln 306 August 3rd

Ln 329 Given that Heinold and Marsham both use the UM (and I don't know what the others used but I suspect not the UM) I think you should comment on the possibility that this is a difference in model physics that is driving the different behaviour.

Ln 364 Once again you are not trying to explain the reason for this. In modelling of convective storms it is a well known phenomena that the radius of updarughts and downdraughts scales with the grid spacing. Could it not just be a similar effect you are seeing here. The same overall vertical motion occurrs but not over such a large area (due to updraught and downdraught scaling with grid spacing) and therefore the average of grid points with non zero vertical wind speeds is relatively higher.

Ln 365-366 This needs to be reworded. At the moment it sounds like you are saying that the mean updraught speeds (throughout the depth of the model) are greater than the mean downdraught speeds near the surface. I suspect what you mean is that near-surface updraughts are greater in magnitude than near-surface downdraughts (would also be nice to give a height blow which this is true).

Ln 395 "in the absence of any" ?

Discussion and reccomendations and Conclusions. Do you really need both sections. There is a good deal of repetition between the two sections straight after one another. I would prefer a single Discussion and conclusions section (afterall, surely recommendations are a conclusion you arrive at from doing the work).
* * *

---

## Author Comment (AC1) · 11 Oct 2019

To begin, the authors would like to thank the reviewer for their time, attention to detail, and insights on the paper and research. Each comment will be addressed point by point. The \* next to line numbers indicates that it is referencing the tracked-changes manuscript. The \* next to figure numbers references the supplemental figures in this response to the reviewer and not the original manuscript.

**Specific Comments:**

**l.175: What is the meaning of 'coupled'? Probably only the use of the mineral dust emissions module (but not the transport, mizing, deposition etc.). Please better explain.**

Here, the term "coupled" indicates that the meteorology (WRF) and the aerosol module (GOCART) are combined in the model in a way that they can directly impact each other. This is not just the meteorology and land surface part of the code being connected to dust emissions (e.g. wind speed, soil moisture, etc.), but also dust transportation via advection, convection, and turbulent mixing, as well as dry / wet deposition, and aerosol radiation effects. The rates for all of these dust processes are inextricably linked to the meteorology, and are treated such in the code via the direct coupling of WRF to the GOCART model. Additional clarification of "coupled" was added to this paragraph for readers:

Ln 154-156\* [The model is coupled to the Goddard Chemistry Aerosol Radiation and Transport (GOCART) module (Ginoux et al., 2001), which allows for feedbacks between the meteorology and aerosols and is described in more detail in Sect. 2.2.]

Ln 184-186\* [...it is then transported based on the simulated meteorological fields from WRF, including advection, convection, and turbulent mixing...]

**I.177 More details are needed about the schemes used. The paper is a sensitivity study about these schemes and they are not explained. In particular, the way to treat the aerosol for the indirect effects is completely different (the Grell scheme is aerosol aware compared to the others).**

There is a paragraph later in the manuscript that points out the major differences between the cumulus schemes tested (Ln 230-240\*). The Grell aerosol-aware scheme mentioned by the reviewer is the Grell– Freitas Ensemble Scheme (Grell & Freitas, 2014), whereas the one tested here is the non-aerosol aware version referred to as the Grell 3D Ensemble Scheme (Grell 1993; Grell & Devenyi, 2002). The aerosol aware scheme was not tested because it depends on the modelled aerosol number concentration affecting the CCN number. However, the GOCART aerosol module is a single-moment in mass scheme, which means it carries no number information and cannot alter the CCN number. As such, the GOCART model wouldn't have an effect on the aerosol aware version.

The overarching point of this study is that resolution matters more than the choice of convective parameterization. Thus, the point of including several cumulus parameterization schemes rather than just one was to represent the uncertainty across a spread of different available options in the model, and not to attribute why one scheme produces one solution or another. That is why at no point in the paper are the cumulus parameterizations directly compared to each other. Rather, they are represented as an ensemble mean with uncertainty estimates. Comparing the detailed responses of individual schemes to each other is outside the scope of this paper, but absolutely warrants further study and could be an entire manuscript on its own merit.

**l.178: for a mineral dust study "no chemistry" why not. But no initial and boundary conditions for a simulation of 3 days, it is not possible to have realistic results**

The only aerosol the authors were interested in for this study was dust. Furthermore, the aerosol burden over the deserts in the Arabian Peninsula is dominated by mineral dust (Heald et al. 2014), and as such, the authors decided that the full atmospheric chemistry code in WRF-Chem (e.g. gas phase and aqueous chemistry, etc.) was not needed and that other aerosol species were outside the scope of this study.

Two additional test cases were performed to address the reviewer's comment relating to the use of initial and lateral boundary conditions. First, a 3 km BMJ simulation was run using both initial conditions (ICs) and boundary conditions (BCs) for dust from the Community Atmosphere Model with Chemistry (CAM-Chem) global model, the output of which can be used for initializing the aerosol and chemistry fields in the mesoscale WRF-Chem model. Second, a 3 km BMJ simulation was run with only the lateral boundary conditions from CAM-Chem.

In the attached supplementary figures (denoted by a \* to differentiate them from the figures in the manuscript), it can be seen that these two test cases (labeled as "BMJ-bcs and ics" and "BMJ-bcs only") have very little effect on the dust uplift potential (Fig. 4\*), the threshold velocity, surface settling flux, bowen ratio (Fig. 5\*), or the mean vertical velocity (Fig. 8A\*). This is expected, since all of these fields are dominated by the meteorology and not the dust concentrations in the local environment.

Furthermore, the second test case "BMJ-bcs only," in which only lateral BCs were used to represent dusty air moving across the domain from places like the Sahara, has essentially no effect on the results and is in line with the conclusions from the manuscript where no lateral BCs were used. Including the BCs has minimal influence on integrated dust (Fig. 6A\*), vertical dust concentrations (Fig. 7A\*), dust fluxes (Fig. 8C\*), or dust radiative effects (Fig. 11\*). There are two possible interpretations of this result. One is that the dust concentrations being transported laterally into the domain are small for this case study. A second possibility is that the CAM-Chem model underestimates this dust source.

Conversely, including dust ICs ("BMJ-bcs and ics") does have an effect on the dust concentrations in the domain. Looking at the integrated dust plot (Fig. 6A\*), there is a decreasing trend starting from the initial timestep throughout the rest of the simulation. The decreasing trend is not seen in the AOD observations from AERONET in the region (Fig 12\*). This points to the ICs being significantly higher than what the model produces on its own, and they are out of sync with the equilibrium model solution. Furthermore, comparing modeled AOD to the AERONET stations, there is little added benefit in using the ICs to get better agreement with observations. The model underpredicts dust throughout the simulation compared to observations, which is a finding in Saleeby et al. 2019, where this particular case study simulation is further compared to observations. It would most likely be better to adjust the dust tuning parameter (C) in Eq. 1 than to use ICs that are not in tune across modeling platforms or in keeping with the observations. The result of the added dust in the IC run is higher integrated dust (Fig. 6A\*), vertical dust (Fig. 7A\*), dust flux (Fig. 8C\*), and a stronger radiative effect. While including dust ICs increases the dust load, it does not however change the conclusions of the study.

Saleeby, S. M., van den Heever, S. C., Bukowski, J., Walker, A. L., Solbrig, J. E., Atwood, S. A., Bian, Q., Kreidenweis, S. M., Wang, Y., Wang, J., and Miller, S. D.: The influence of simulated surface dust lofting and atmospheric loading on radiative forcing, Atmos. Chem. Phys., 19, 10279–10301, https://doi.org/10.5194/acp-19-10279-2019, 2019.

**I.182 "kept constant" meaning remain the same during the whole simulation?**

Correct, and that these physics options don't change across the simulations – the language has been changed in the manuscript to remove any confusion:

Ln 162\* [The following model parameterizations were employed and kept constant across the simulations..]

*I.214: The 'Dust Uplift Potential' is a calculation already done in a large majority of dust emissions schemes, by principle of the mechanism to evaluate. Unfortunately, it represents only a small part of the problem and is not really useful. It describes only the link between the friction velocity treshold (using the aeolian roughness length) and the current friction velocity. But other important parameters are not taken into account: the vegetation, the erodibility, the soil humidity, the recent precipitation etc. In addition, the fact to use a constant Ut is not realistic (eq.3): the aeolian roughness length is far to be constant over erodible region. It is the most important varying parameter in mineral dust emissions modelling. The use of three different kind of DUP has a large interest. The message is already contained in one. If the authors really want to use this criteria, only one is enough.*

The authors are not sure exactly as to what is being asked here by the reviewer. However, we have done our best to address the questions here as we understand them and hope this will address the reviewer's concern.

Several of the parameters listed here are contained in the varying DUP equations, including the erodibility (Eq. 5 with the variable S) and the soil moisture / recent precipitation (Eq. 4 and 5 depend on  $U_t$  - the only varying parameter in Eq. 2 for  $U_t$  is soil wetness,  $w_{soil}$ ). The point of including these different DUP parameters is to tease out which of these processes is the most important without assuming that one is more important than the other for this case study. It has been shown previously in the literature that the soil moisture and erodibility are important for dust uplift (i.e. Gherboudj et al. 2015) in addition to wind speed, which means that using only one parameter doesn't tell the whole story.

Eq. 3, which is the most simplistic of the equations and assumes a constant roughness length, and has been used widely in the literature, especially in offline model dust approximations. To compare our results with that of other studies, it is necessary that we use Eq. 3. However, we point out its limitations and have included the more complicated DUP parameters we think are more useful for this study:

Ln 205-207\* [This simplified equation for dust uplift has been used in previous dust studies, and is useful to include here to place the findings of this manuscript in the context of existing literature.]

*1.239: The reference simulation has an horizontal resolution of 3km to enable explicit convection calculation. This simulation has boundary conditions and this is a good point. But these boundary conditions are from the BMJ simulation, i.e one of the studied case. Thus, we can think that the reference case will be very influenced by this case, no? To have a more realistic comparison between scheme, the 'reference' has to be done for each scheme and a first spread can be calculated between all 'high resolution' cases.*

This is a good point – especially since there are two competing classes of cumulus parameterizations tested here: BMJ is the only moisture / temperature adjustment scheme, whereas the others are mass flux schemes. To test the sensitivity of the results to which cumulus parameterization scheme is employed in the parent nest, a second 3 km simulation was run. In this test, the Kain-Fritsch cumulus scheme serves as the 15 km parent, which is then nested to 3 km (this run is labeled as "3 km – KF" throughout the supplementary figures) to represent the mass-flux schemes. In none of the figures (Fig. 4-11\*) is this difference significant or does it change the conclusions of this paper. Again, because model resolution dominates over the choice of cumulus parameterization, this effect of using a different parameterization in the outer nest has little effect on the results.

Ln 227-228\* [Other combinations of nests were tested, but the results were not sensitive to which 15 km simulation was used as the parent nest, or which lateral boundary conditions, for the 3 km simulation.]

**l.263: for long-range transport, 24h of spin-up is not enough. For the time averaged results, it is only th elast two days. But for the time series, it is the 3 days? why this difference?**

For the long-range transport part of this comment – see the response and tests from comment I.178. Including the lateral boundary conditions that represent long-range dust transport have little impact on the results and hence we feel that 24h of spin-up is sufficient.

For the time averaged results, we did not want to include the 24-hour model spin up time. However, they were included in the time series to show how the model approaches its equilibrium solution when starting from no dust sources.

**I.272: why not use directly the mineral dust emissions fluxes? Please explain this important point.**

The emission fluxes convey the same story as the approach utilized here (see Fig. 13\*). However, the difference in the magnitude across the simulations is difficult to see in this plot, and it wasn't included. The conclusions are the same from a dust emission standpoint versus a dust concentration perspective. Additionally, the variables that go into the emission flux formula in GOCART (Eq. 1) are very similar to the Dust Uplift Potential (DUP) calculations. The most complete DUP parameter (Eq. 5) includes the same variables as the emission flux, so it would be redundant to include both the DUP calculations and the emission fluxes.

**I.280: why the simulation with the coarsest resolution (and not simulation) overestimates the wind speed? Please explain (and I imagine it is the "10-m wind speed", please correct).**

Correct – it is the 10-m wind speed. This has been updated in the manuscript:

Ln 297-298\* [The coarsest simulation overestimates the near-surface wind speeds related to the NLLJ mechanism, which...]

There are a few theories regarding why the coarsest simulation would overestimate the near-surface wind speed. Marsham et al. (2011) noted that in their simulations over Northern Africa, the Saharan Heat Low was more pronounced in the coarse simulations. They postulated that cold pool venting in the explicit simulations reduced this thermal low, thereby reducing the horizontal pressure gradients which are responsible for low-level jets in this region. It follows then that the low-level jet is weaker in this

scenario, as is the process of mixing of the jet to the surface, and this the near-surface wind speeds. In the Arabian Peninsula case study, this mechanism is certainly quite possible. However, this theory has yet to be tested and is outside the scope of this paper.

*I.293: Yes, it is right. And obvious. Of course, a key point in modelling is to try to have a model not sensitive to the spatial resolution. And it seems it is the problem with WRF-chem. In WRF, the principle is to use, for each grid cell, the dominant soil type and landuse. Thus, by principle, the result is very sensitive to the resolution. Some other models are using subgrid scale variability and Weibull distribution for the 10-m wind speed, for example, to avoid this problem. Please see bibliography and replace WRF-chem in the context of all currently used regional dust models.*

WRF-Chem is just one of many regional models that can be used operationally and / or in research applications. For instance, The Sand and Dust Storm Warning Advisory and Assessment System (SDS-WAS) includes 12 dust models, with more undoubtedly available to be used in research applications. Each model is unique, and most likely has several options for their cumulus parameterizations as well as other physical representations of meteorological processes. Combining the differences between dust models in this way is a very large undertaking the likes of which are being conducted by organized working groups like the International Cooperative for Aerosol Prediction (ICAP) and is outside the capabilities of a single manuscript.

*I.331: it is not sure that there is an interest to have a conclusion such as "resolution increases or decreases the mineral dust emission fluxes". In fact it depends on the studied area, the variability of the orography, aeolian roughness length, soil humidity, vegetation. And, of course, the way to well take into account or not all these processes and their variability.*

The reviewer makes an excellent point here. The manuscript has been updated to include more about the uncertainty here:

Ln 355-360\* [...dust emissions and airborne dust mass increases in the WRF-Chem simulations in the convection-allowing simulation, which is in closer agreement to the studies of Reinfried et al. (2009) and Bouet et al. (2012) who used COSMO-MUSCAT and RAMS-DPM respectively. Considering each study used a different model and therefore physics, it is unsurprising that the results vary. However, it is not apparent how much of a role the region or specific case study plays in this difference and is an area for future work. ]

*I.335: I don't understand the discussion with "The rates of gravitational settling are higher in the explicit simulation compared to the coarse simulations, yet Fig. 6.a suggests that this is not enough to offset the higher dust emissions, or the integrated dust quantities would be similar across all the simulations." The dry deposition is proportional to the concentrations, being a velocity applied to the concentrations. How is it possible to have 'enough' settling to 'offset' the higher dust emissions?*

If there is more dust aloft, more dust eventually needs to settle out. The point that we are trying to make here is that the missing piece in this process is the higher vertical transport. If dust was transported to the same height, the gravitational settling would offset the higher emissions and there would be no reason for the integrated dust values to be higher. This part of the manuscript has been edited for clarity.

Ln 364-367\* [The rates of gravitational settling are higher in the convection-permitting simulation compared to the coarse simulations because more dust is available aloft to settle out. Nevertheless, Fig. 6.a suggests that this increase in gravitational settling rates in the 3 km case is not enough to offset the higher dust emissions...]

Figure 6: the fact to have difference sbetween resolution is understandable but a factor 2 has to be better explained. Mineral dust emissions mass maps for the common domain (the one with 3km horizontal resolution). The caption is not easy to understand: "Domain averaged integrated dust mass". Please correct with Spatially averaged, vertically integrated.

The difference between resolutions in Figure 6 differ by a factor of 1.5, which we discuss in the previous section with Figure 4 and Figure 5. Using DUP(U,Ut,S) we see that the 3 km has the most potential to loft dust, especially on 04-Aug when there is a convective maximum. This is related to the threshold velocity being lower and soil wetness (Figure 5) and is also explained with the differences in vertical transport, which is covered in the next section of the manuscript. More about the differences in precipitation in convection-allowing versus parameterized simulations affecting soil moisture and the threshold velocity has been included in the text:

Ln 308-312\* [Rainfall is generated differently in parameterized versus convection-allowing simulations, and it has been well documented that parameterized simulations produce more widespread light rainfall, whereas more intense rainfall tends to develop over smaller areas in convection-allowing simulations (e.g. Sun et al., 2006; Stephens et al., 2010). From a domain average perspective, rainfall in the 3 km simulation will cover less area, leading to the soil moisture threshold not being exceeded as frequently compared to the parameterized cases.]

These figures have been updated for clarity and the captions have been changed. Throughout the manuscript anytime there is a reference to "domain averaged integrated dust" it has been changed to the phrase "spatially averaged, vertically integrated."

Ln 911\* [Figure 6: Spatially averaged, vertically integrated dust mass. Colors and shading are identical to that in previous figures.]

*I.346: "the vertical dust profile follows a generally exponentially decreasing function" is it a conclusion of this study? or coming from a reference? These is no reason to have an exponential decrease in the troposphere. Many cases of thin but concentrated dust plumes transports are observed and modelled...*

On average, exponentially decreasing aerosol in the troposphere is a good assumption (e.g. Gras 1991; Tomasi, 1982). This type of idealized profile is often assumed for CCN in models (e.g. Fan et al., 2007). You are correct in that individual plumes will change this profile, but here we are looking at a domain average, which regresses to the exponentially decreasing function.

*I.369: "The implications for dust transport based on vertical velocities is convoluted." This sentence is difficult to understand.*

This part has been further explained in the text to avoid confusion:

Ln 403-404\* [The implication for dust transport based on vertical velocities is convoluted, since updrafts and downdrafts work concurrently to redistribute aerosol.]

**I.421: The impact on radiation, with potential heating and cooling, is a process needing more than 2 days of simulation to be significative.**

The timescales of interest vary depending on which specific processes are being examined. From a climate perspective, two days is much too short. However, looking at static stability in the lower atmosphere from a mesoscale perspective, including processes like convective initiation or the formation and deterioration of the nocturnal low-level jet, the timescales examined here (or in some cases even shorter timescales) are important and significant.

**I.428: there is a sign change. Could you explain why?**

The model applies a higher weight (via the refractive index for mineral dust) to dust scattering in the shortwave and cooling compared to the longwave absorption. With more dust in the explicit case, the shortwave effect is amplified.

More explanation has been added to this section for clarity:

Ln 468-470\* [The model has a stronger shortwave effect for dust based on the prescribed index of refraction, but is also related to the timing of dust emissions, considering the SW effect is only active during the daytime.]

**General Comments**

1. There is no data used in this work: the simulations are compared between them but we have no idea of the realism of the simulations (there is only one reference for a comparison to Aeronet AOD in another paper, under discussion, and no guarantee this is exactly the same model set-up, and which one?). At least, the reference case (dx=3km) should be compared to available data (surface networks such as MIDAS, AERONET, satellite, other data).

The Saleeby et al. 2019 study (referenced above and in the paper) where the 3 km simulation was compared more thoroughly to observations has been published (once again provide the full reference here). In that paper, the exact same model setup was used for the 3 km simulation as was used here, and this point has been added to the manuscript. We have included comparison to the few AERONET sites in this region in the supplementary Fig. 12\*, and found similar results (regardless of including or excluding the ICs and BCs in the simulations) with Saleeby et al. (2019) in that WRF-Chem under predicts AOD. However, the model must assume a refractive index for dust to calculate AOD, which may or may not be realistic in itself. Additionally, we have selected dust as the only aerosol present in the model, while in reality there are other aerosol types that may be contributing to the AOD. Thus, making one-to-one comparisons here with observations is difficult. None of the continuous observational networks provide dust concentration, which is what is actually needed for a true validation. Nevertheless, if WRF-Chem is underpredicting dust concentrations, this doesn't change the conclusions of the study.

2. The studied case extended from 2 to 5 August 2016: there is no spin-up time, important when studying transport of aerosol such as mineral dust. Time series are presented for the three days, but some average are done only for the last two days, explaing that the first day is spin-up. But, viewing the domain size, the minimum spinup time should be at least one week.

See response to comment on I.178 above.

3. There is no boundary or initial conditions. These missing background values may have a large impact on the results, in particular knowing that the model couples the meteorology and the aerosol concentrations: direct and indirect aerosol effect may be long-term and it is required to have correct boundary conditions to have realistic effect of aerosol on meteorology. For the 'reference' domain, the boundary conditions are extracted from one of the studied case, biasing the results.

See response to comment on I.178 above.

4. The convection schemes used are not explained. The paper is a sensitivity study about these schemes but there is no explanations about their real differences, how they take into account aerosol or not, thus no conclusion about why results may be different depending on the scheme.

**See response to comment I.177 above.**

5. The paper deals with the sensitivity to the model resolution. But since the schemes are not well implemented (no wind speed distribution, no subgrid scale variability), there is a large sensitivity but not for realistic and physical reasons: the differences are not due to the convection schemes in general but just to the fact that the problem of the resolution is not well designed in this model: it is not possible to describe a threshold problem (such as mineral dust emissions) without taken into account disstributions of input parameters. Results are linked to this model only and are not useful for other modellers

Regardless of how successfully or unsuccessfully these schemes have been implemented into WRF-Chem, it is still a very widely used model for air quality, atmospheric chemistry, and more relevant for our manuscript - dust research and forecasting. A list of some of the current forecasting centers using WRF-Chem can be found on the WRF-Chem users page (https://ruc.noaa.gov/wrf/wrfchem/Real\_time\_forecasts.htm) and is one of the dust models included and evaluated in the SDS-WAS real-time forecasts.

WRF-Chem users need to be aware of its limitations and its sensitivity to resolution when designing numerical experiments, and readers should be cognizant of this when interpreting results from both past and future studies that use this model. Furthermore, some of the results we found here are similar to other studies that have used different regional models, such as Reinfried et al. (2009), while other manuscripts are in disagreement with our findings, such as Heinhold et al. (2013) and Marsham et al. (2011). Clearly, we have not reached a consensus and more work is needed. Between the user base for the WRF-Chem model and the spread in results between our findings and previous literature, there is a broader community of interest for this paper.

Supplementary Figure 3) Same as in Fig. 3 in the manuscript, but the location of the 3 AERONET sites in the analysis have been added.

---

## Author Comment (AC2) · 11 Oct 2019

Thank you to the reviewer for their insights on the paper and research. We believe the manuscript is stronger based on their comments and we are appreciative of the thoughtful advice included here. Each comment will be addressed point by point. The * will denote line numbers in the tracked-changes manuscript.

**General Comments:**

*I know that it is common parlance in the community to refer to simulations that are run without the use of convective parameterizations as being "explicit" or that convection is is "explicitly represented". However, more recently there has been a shift towards the use of simulations of this type being referred to as "convection permitting". This difference is subtle but I think is a better descriptor of what the models are actually doing. The model grid-scales involved are not so fine as to explicitly resolve individual updraughts and downdraughts but are sufficiently high to permit the development of convective storms that approximate those that we might observe in reality. I feel that it would be better to replace descriptions of simulations currently described as explicit with convection permitting.*

This is a very good point. We agree that the term "convection permitting" is a more accurate description of the representation of convection in the model compared "explicit." Like the community, we have made the mistake of equalizing the two terms in the manuscript, when really only "convection permitting" should be used. The manuscript has been updated to replace "explicit" where possible with the terms "convection-permitting" and "convection-allowing."

*Did you consider running a 15 km simulation with the convective parameterization switched off. I don't think that you should do this as the work is already of a high standard, but think that you might well be surprised at how small the difference is between a 15 km grid-spaced convection permitting simulation and a 3 km grid-spaced convection permitting simulation*

We actually did run a 15 km simulation without a convective parameterization, but decided not to include the results in the manuscript. The spatial and time averaged results from the no parameterization case are, in fact, similar in magnitude to running at 15 km with a cumulus parameterization. Differences do occur in the timing of the different local dust maxima throughout the day. This points again, to resolution being the dominate factor to control in this simulation rather than the choice of cumulus parameterization (or the choice to even employ a cumulus parameterization at that grid spacing at all). A short discussion of this has been added to the manuscript:

> Ln 223-224* [A 15 km simulation with no cumulus parameterization was also tested, but the results were similar and within the spread of the 15 km simulations that employed cumulus parameterizations and are not included here.]

**Specific Comments:**

*Ln 17 -20 You need to be clear that the updraughts that are transporting dust vertically are part of the general circulation (eddies) in the dry atmosphere. At first I thought you were specifically talking about storm updraughts (which I assume are less important in the simulation for vertical dust transport due to washout).*

It's a combination of both, but yes, the storm updrafts are mediated by wet deposition, whereas the dry eddies are not. This point has been included:

Ln 67-69* [Current aerosol forecast and climate models are run at fine enough grid-spacing to simulate synoptic events but still typically employ cumulus parameterizations, which are incapable of resolving dry and moist mesoscale updrafts and downdrafts that can potentially loft and / or scavenge dust.]

*Ln 45-47 I think it would be wise to indicate that in reality ingestion of this type is impossible. What you are hoping for is that the initialisation data and the representation of dust are good enough for your purposes. It is perfectly possible that that is true for this case study but that the same setup run for different case studies could provide different results due to the high dependency of models (even those that do not contain dust) on initial conditions.*

We agree with the reviewer's point. The spread across models (and within the same model based on physics options) can be vast. More has been included in this section to emphasize the limitations here based on model and case study choice:

Ln 47-49* [Even the state-of-the-art models are currently incapable of this type of assimilation and rely on the quality of the dust model and initialization data, which models are known to be especially sensitive to and will vary depending on the specific region and case study.]

*Ln 47-49 Is it the global and regional nature of models that causes these differences or is it the grid-spacing or other model differences? Please be clear.*

The dust model inter-comparison studies listed in the text varied in terms of grid resolution (horizontal and vertical) and model physics (including the dust schemes), even for the same case study. However, the grid resolution of the models was consistent in that they were all at grid-spacings that would employ a cumulus parameterization. The literature referenced here was not comparing global versus regional, but if those studies exist we are interested to see the results. The text has been updated to reduce confusion here:

Ln 49-52* [As such, substantial discrepancies exist across global models of similar resolution (Huneeus et al., 2011), and across regional models (Uno et al., 2006; Todd et al., 2008) in the magnitude of predicted dust flux from the surface to the atmosphere, as well as the models' overall representation of the dust cycle.]

*Ln 50 I would get rid of "accurately" here. Generally in models dust processes are fairly simplistic and highly parameterised and so the idea that dust processes are accurately represented is a fallacy.*

True, it's a stretch to say that the highly parameterized physics in the model could be thought of as "accurate". The word "accurately" has been removed from this and the next section.

*Ln 53-59 This section needs rewording. The first sentence along with the word "Additionally" suggests that large-scale, synoptic-scale and meso-scale meteorology is separate from the phenomena listed below. Also why say large and synoptic scales? Instead I would suggest something like "Dust uplift events can be associated with meteorological processes across a broad range of scales. Synoptic scale uplift phenomena include monsoon troughs (Marsham et al, Beegum et al), Shamal winds (Yu et al.) and*

*frontal systems (Beegum et al). While dyamical effects on smaller (meso) scales can raise dust through the production of convective outflow boundaries (haboobs; Miller et al.) and the morning mixing of nocturnal low level jet (NLLJ) momentum to the surface (Fiedler et al)."*

Thank you for the clarification. The wording suggested by the reviewer is a welcomed improvement and has been included in the text:

> Ln 57-61* [Synoptic scale uplift phenomena include monsoon troughs (e.g. Marsham et al., 2008), Shamal winds (e.g. Yu et al., 2015) and frontal systems (e.g. Beegum et al. 2018), while dynamical effects on smaller (meso) scales can raise dust through the production of convective outflow boundaries, or haboobs, (e.g. Miller et al. 2008), daytime turbulence or dry convective processes (e.g. Klose and Shao, 2012), and the morning mixing of nocturnal low level jet (NLLJ) momentum to the surface (e.g. Fiedler et al. 2013).]

*Ln 60 What other drivers of dust emission are there? There are prerequisite conditions (dry, unvegetated surface etc.) but wind is the only driver of surface dust emission that I can think of.*

Possibly some anthropogenic activities can emit dust (e.g. plowing agricultural fields, construction, etc.), but ultimately, it's still then transported away from the source by the wind. This line was replaced to point out that wind is the only driver (albeit modulated by other conditions) and that we are only considering meteorological processes here:

> Ln 61-62* [When considering only meteorological dust sources to the atmosphere, wind drives dust emissions…]

*Ln 73 Heinold used offline emission which I think is a relevant point to mention here as it significantly differs from your approach. Another paper that discusses the grid-scale effects on online model dust and convective representation of dust in West Africa would be Roberts et al. 2018 (doi.org/10.5194/acp-18-9025-2018).*

Yes, that is definitely worth mentioning and has been included. It's an important point for understanding the importance of the DUP parameter in the context of other studies. The Roberts et al. 2018 paper has also been added to the literature review to better place our results in the context of existing literature:

> Ln 78-79* [Heinold et al. (2013) ran the UK Met Office Unified Model (UM) over West Africa with offline dust emissions, and found that…]

> Ln 86--88* [Roberts et al. 2018 also used UM to investigate this relationship over the Sahara and Sahel and reported little change in the dust emissions when moving from parameterized to explicit convection, but also noted that the NLLJ maximum decreased as theconvective maximum increased.]

*Ln 82 One thing that you don't mention is that the thing that effects models the most is not the grid scale, or the microphysics and in some cases not even the whether simulations are convection permitting or parameterized. It is the initialisation data. This is one of the findings in Schepanski et al. 2015 (doi.org/10.1002/qj.2453) in West Africa.*

Naturally, the model initialization data are going to be either a substantial source of error or accuracy in the output data. We have added this note and reference to the manuscript to remind readers that the findings here will be modulated by the initialization data:

> Ln 70-73* [Schepanski et al. 2015 found that online dust models are likely to be most sensitive to the initialization data compared to other model options, model sensitivity to the representation of convection will be an additional source of uncertainty in dust forecasts.]

*Ln 104-114 Roberts et al. 2016 (mentioned above) covers some of these points by using the Met Office Unified Model over West Africa. In the UM over summertime West Africa at least, the grid spacing does very little compared to representation of convection.*

These findings have been added to the text (see above comment). But, despite the model and the region being different between these studies, we have found similar results.

*Section 2.1 I find the ordering here a little odd. I would normally expect the model description to precede the description of the conditions that caused the dust uplift. It feels a little like you are skipping backwards and forwards between results and methods. I advise moving your current section 2.1 to either the end of section 2 or the start of section 3.*

The case study description has been moved to the end of Section 2.

*Ln 144-145 I don't think that Figs 1 and 2 show this. The first shows a number of different fields (not dust) and I wouldn't describe Figure 1 as the meteorological setup either. Figure 2 is actually 2 profiles which doesn't match the description either. Please be much clearer in you description. I cannot tell what you are referring to.*

Thank you for pointing this out. We agree that it's more like a snapshot of the meteorology than an analysis of the meteorological setup. A more in-depth meteorological analysis of this case study and an attribution of the dust to different meteorological sources can be found in Miller et al. (2019) and we have directed readers there if they are interested:

> Ln 258-262* [A meteorological analysis of this event, including an attribution of specific dust sources to meteorological features can be found in Miller et al., 2019 and will not be reiterated in detail here. Rather, a snapshot of the meteorology and dust fields from the WRF-Chem simulation on August 3$^{rd}$ at 15:00:00 UTC can be found in Fig. 1-2 as a reference to the typical meteorological setup for this case study.]

*Ln 180-187 A very brief description of why these parameterizations were chosen would be welcome. For instance is this a replication of a setup used in a similar study? Is it similar to operational setups of WRF that are run for similarly arid regions? Or is there an individual reason for having chosen each of these options.*

A reference was added to point out that similar WRF physics options have been used in dust studies in this region:

Ln 163-164* [The following model parameterizations were employed and kept constant across the simulations, with similar WRF physics options being utilized elsewhere to study dust effects (e.g. Alizadeh Choobari et al. 2013:)]

*Ln 289 You should say why the soil moisture is more likely to fall below the threshold in the convection permitting simulations. This is very likely associated with the different way in which rainfall in generated in parameterised and convection permitting simulations. Parameterized simulations have much more widespread light rainfall while convection permitting simulations have rainfall over much smaller areas but at much higher rates. The smaller areal coverage of rainfall in the convection permitting simulations is most probably the cause of the soil moisture threshold not being exceeded as frequently.*

The comment about rainfall affecting the soil moisture is on point. Thank you for raising it. We had similar ideas about this mechanism and have expanded this section to discuss these processes more.

Ln 308-313 [Rainfall is generated differently in parameterized versus convection-allowing simulations, and it has been well documented that parameterized simulations produce more widespread light rainfall, whereas more intense rainfall tends to develop over smaller areas in convection-allowing simulations (e.g. Sun et al., 2006; Stephens et al., 2010). From a domain average perspective, rainfall in the 3 km simulation will cover less area, leading to the soil moisture threshold not being exceeded as frequently compared to the parameterized cases.]

*Ln 306 August 3rd*

Typo has been corrected.

*Ln 329 Given that Heinold and Marsham both use the UM (and I don't know what the others used but I suspect not the UM) I think you should comment on the possibility that this is a difference in model physics that is driving the different behaviour.*

We have added this point throughout the manuscript to remind readers to be cognizant that the models are different and have different physics.

Ln 358-361* […who used COSMO-MUSCAT and RAMS-DPM respectively. Considering each study used a different model and therefore physics, it is unsurprising that the results vary. However, it is not apparent how much of a role the region or specific case study plays in this difference, and is an area for future work.]

*Ln 364 Once again you are not trying to explain the reason for this. In modelling of convective storms it is a well known phenomena that the radius of updarughts and downdraughts scales with the grid spacing. Could it not just be a similar effect you are seeing here. The same overall vertical motion occurrs but not over such a large area (due to updraught and downdraught scaling with grid spacing) and therefore the average of grid points with non zero vertical wind speeds is relatively higher.*

We agree that the scaling of the updraft / downdraft radius with grid spacing is well-known, and this is most definitely a factor here. But pushing this argument further, the finer grid spacing could permit points with higher, lower, or near-zero vertical velocities compared to the coarse spacing. The average does not necessarily have to skew higher and without testing we wouldn't know how that plays out. In

this case, the results skew to higher velocities, which is evident in the CFADs (Fig. 9). We are more likely to witness higher vertical velocities rather than lower or near-zero velocities in the 3 km simulation compared to the coarse simulations. These discussion points have been added to the section.

> Ln 398-400* [It is known that in numerical models, the updraft radius scales with the grid spacing (e.g. Bryan and Morrison, 2012), with a compensating increase in updraft speed as the radius decreases. This relationship skews the frequency of vertical velocities to higher values.]

*Ln 365-366 This needs to be reworded. At the moment it sounds like you are saying that the mean updraught speeds (throughout the depth of the model) are greater than the mean downdraught speeds near the surface. I suspect what you mean is that nearsurface updraughts are greater in magnitude than near-surface downdraughts (would also be nice to give a height blow which this is true).*

Your interpretation is correct – the text has been updated to remove this confusion.

> Ln 400-402* [Irrespective of resolution, the mean updraft speeds in the WRF-Chem simulations are slightly higher than the downdraft speeds, while at the surface mean downdraft speeds are higher than updraft speeds…]

*Ln 395 "in the absence of any" ?*

*Discussion and reccomendations and Conclusions. Do you really need both sections. There is a good deal of repetition between the two sections straight after one another. I would prefer a single Discussion and conclusions section (afterall, surely recommendations are a conclusion you arrive at from doing the work).*

After considering this point, we decided to keep the sections as is and leave the result section more quantitative, with the discussion being more qualitative.

---

## Author Response (AR2)

**Response to Editor**

1) *Provide your thoughts on reviewer's comment #1, and incorporate them into the discussion section.*

Thank you for this suggestion. More about comparing the results to observations was included – and we point out that when compared with observations the model under-predicts dust AOD.

   Ln *289-291 - [More information on model validation of this study, including comparisons of these simulations with AOD observations can be found in Saleeby et al. (2019), which shows that WRF-Chem systematically underestimates dust AOD for this event.]

   Ln *516-518 - [This factor would have to be scaled further, since comparison of the WRF-Chem model to AERONET sites and other AOD observations (Saleeby et al. 2019) shows that WRF-Chem underestimates dust under these conditions.]

2) *While it is not unusual to start all zeros in BC/IC for the aerosol field, this practice seems more appropriate when analyzing results for runs with a longer time period. Based on the figure you showed, results between runs with/without boundary condition/initial condition are not that different on Day 3 but are different enough on Day 2. I don't think your general conclusions will change but do worry that your quantitative results might change because you did include Day 2 in your analyses. The quantitative results matter if you aim to discuss the sign of the model response. Therefore, please carefully go through your analyses, and update your results accordingly. If indeed nothing significant changes, please do use a few statements to inform readers about this part of work.*

The supplemental figures in the response to RC-2 have been double-checked and the choice to use BC/IC's does not change the conclusions of this study. The additional dust in the domain at the beginning of the simulation enhances the dust radiative effect (Supplementary Figure 11), but this doesn't change the timing of the processes in the dust uplift parameter (Supplementary Figures 4A, 4C, 4E) because those values are dominated by the meteorology and surface characteristics, and much less so by the airborne concentration of dust. More about the BC/ICs has been added to section 2.1 concerning the model setup.

   Ln *145-154 - [The model was tested with and without dust initial and boundary conditions from the Community Atmosphere Model with Chemistry (CAM-Chem) global model (Emmons et al. 2010). The concentrations of dust advected through the lateral boundary conditions was too small to have an effect on the results, and the initial conditions introduced a spurious decreasing integrated dust trend over time when modeled aerosol optical depth (AOD) was compared to AERONET observations. While the initial conditions led to a higher integrated dust mass, it did not change the conclusions of the study. To remove this factor and focus more on the meteorological processes that actively loft and transport dust in real-time, no chemistry or aerosol initial / lateral boundary conditions were used. Rather, the aerosol fields were initialized with zero concentrations and were allowed to evolve naturally from the model meteorology, aerosol, surface and radiation processes.]

3) *Regarding Reviewer's comment #4, please briefly summarize the evaluation results in a couple of sentences.*

More was added to section 2.3 about the cumulus schemes and 2.3 about how aerosol cannot feedback through the microphysics to the cumulus scheme via indirect effects.

Ln *169-171 [GOCART is single-moment in mass, meaning there is no number information available to change the number of cloud condensation nuclei or ice nuclei in the microphysics. As such, the indirect effects of dust cannot be simulated with this setup. Through this model, dust is emitted to the atmosphere in 5 discrete effective radii bins…]

Ln *241-245 [Several cumulus parameterization schemes were tested to introduce spread into the solutions and to represent the 15 km results as a 5-member ensemble mean with uncertainty estimates. Because this paper seeks to investigate the effect of horizontal resolution on dust transport, comparing individual cumulus schemes against one another is outside the scope of this paper.]

4) *Could you please make introduction more concise and get to the point? The current form is a very nice literature review, but some of work cited there does not closely link to the aims of the manuscript. I am hoping that a more concise introduction can help to highlight the scientific significance of the manuscript more clearly.*

And effort has been made to shorten each paragraph slightly, and a few less-relevant citations have been removed (see tracked changes document for deletions). The authors feel that what remains in the introduction is vital to framing the background of this project and is necessary to address some of the points raised in the reviewer comments, especially those from RC2 who pointed out some missing citations and background information that should be included.

[revised manuscript text omitted]

---

## Author Response (AR3)

**Response to Editor – Round 2**

*Thanks for your detailed responses and revisions. The manuscript is accepted, but please do make one more very minor change before you submit the final version. Please put line 64-67 back to your introduction, since it is important to mention that.*

Thanks for the input – you're correct in that these lines are important. They have been added back into the revised manuscript.

[revised manuscript text omitted]